# Venous Thromboembolism in Cancer Patients Undergoing Chemotherapy: A Systematic Review and Meta-Analysis

**DOI:** 10.3390/diagnostics12122954

**Published:** 2022-11-25

**Authors:** Ming-Yee Sun, Sonu M. M. Bhaskar

**Affiliations:** 1Global Health Neurology Lab, Sydney, NSW 2000, Australia; 2Neurovascular Imaging Laboratory, Clinical Sciences Stream, Ingham Institute for Applied Medical Research, Liverpool, NSW 2170, Australia; 3UNSW Medicine and Health, University of New South Wales (UNSW), South Western Sydney Clinical Campuses, Sydney, NSW 2170, Australia; 4Department of Neurology & Neurophysiology, Liverpool Hospital & South West Sydney Local Health District (SWSLHD), Liverpool, NSW 2170, Australia; 5NSW Brain Clot Bank, NSW Health Pathology, Sydney, NSW 2170, Australia; 6Stroke & Neurology Research Group, Ingham Institute for Applied Medical Research, Liverpool, NSW 2170, Australia

**Keywords:** cancer, chemotherapy, venous thromboembolism, prevalence, screening

## Abstract

**Objective**: Venous thromboembolism (VTE) is a life-threatening complication that may exacerbate cancer prognosis. Whilst some studies indicate an increased risk of VTE in cancer patients undergoing chemotherapy, the prevalence estimates on the pooled prevalence of VTE in cancer patients undergoing chemotherapy are not known. This study aims to calculate the pooled prevalence of VTE in chemotherapy-treated cancer patients. **Methods**: Studies on VTE occurrence in cancer patients undergoing chemotherapy were retrieved after database search. The terms used included “cancer”, “chemotherapy”, and “venous thromboembolism”. A random-effects meta-analysis was conducted to obtain a pooled estimate of VTE prevalence in cancer patients undergoing chemotherapy. **Results**: A total of 102 eligible studies involving 30,671 patients (1773 with VTE, 28,898 without) were included in the meta-analysis. The pooled estimate of VTE prevalence was found to be 6%, ranging from 6% to 7% (ES 6%; 95% CI 6–7%; z = 18.53; *p* < 0.001). **Conclusions**: The estimated pooled prevalence rate of VTEs was 6% in cancer patients undergoing CRT, which was higher than the overall crude prevalence rate (5.78%). Comprehensive cancer care should consider stratified VTE risk assessment based on cancer phenotype, given that certain phenotypes of cancer such as bladder, gastric and ovarian posing particularly high risks of VTE.

## 1. Introduction

Venous thromboembolism (VTE) is a major public health problem constituting a significant burden of disease [1,2]. There are around 10 million cases of VTE worldwide every year. After myocardial infarction and stroke, VTE is the third leading vascular disease [3]. In the first one to three months following a stroke, there is an increased risk of VTE, partly because of immobility brought on by the stroke [4]. Major venous and arterial thrombotic disorders share overlap in some key cardiovascular risk factors [5]. A higher risk of VTE is linked to specific cardiovascular risk factors such as older age, smoking, and greater adiposity [2,6]. Cancer, a leading cause of death and disability in the world [7,8], is known to potentiate the risk of VTE and roughly 20% of VTE are linked to cancer [9,10]. Thrombosis in cancer patients is a clinically challenging construct which is associated with poor outcomes despite therapy [11].

Recent studies have also indicated that cancer patients on chemotherapy may be at an increased risk of venous thromboembolism [12,13,14]. Assessment of VTE risk is critical for appropriate medical management and prophylactic treatment [15]. Given the lack of data on the prevalence estimates of VTE in cancer patients, especially in those receiving chemotherapy, further studies are required. Distinct cancer phenotypes may render cancer patients at varying levels of VTE risk [16,17,18]. Recent guidelines from American Society of Haematology published in early 2021 recommend stratifying cancer patients according to their VTE risk prior to the start of chemotherapy, as well as patient-specific factors, using the Khorana risk score, the major determinant of which is cancer phenotype [19]. This comes in the background of two landmark randomized clinical trials (RCTs), resulting in the change of guidelines, demonstrating VTE prophylaxis with direct oral anticoagulants (DOACs) following risk assessment lowered the incidence of VTE during chemotherapy [20,21,22]. Several societies or health systems beyond United States are yet to adopt these recommendations; besides, unwarranted variations in clinical care as well as poor adherence to recommendations or guidance vis à vis VTE risk assessment and optimal administration of thromboprophylaxis pose an ongoing real-world or systems challenge [23,24]. Moreover, literature is sparse when comparing the relative risk and prevalence of VTE across multiple cancer phenotypes—with studies only revealing VTE prevalence specific to a cancer phenotype and risk in homogenous cancer populations, vis à vis their ethnicity and treatment received. Understanding of, and estimates of, the pooled prevalence may also be useful to increase awareness on VTE risks in cancer patients undergoing chemotherapy as well to inform clinicians and patients on the quantum of the VTE prevalence/risks in cancer or across various types of cancer. This meta-analysis sought to investigate the pooled prevalence of venous thromboembolism in cancer patients receiving chemotherapy. There is also a gap in clinician knowledge pertaining to the specific risk that cancer phenotypes and chemotherapy poses to cancer patients. We have sought to address two key underlying questions through this meta-analysis:(1)what is the prevalence of VTE in cancer patients receiving chemotherapy?(2)what is the prevalence of VTE stratified by cancer phenotype in patients undergoing chemotherapy?

## 2. Materials and Methods

### 2.1. Literature Search: Identification and Selection of Studies

The primary search engine of this meta-analysis and systematic review was the PubMed database. Articles published between 2012 and October 2022 were included in the search. Search terms included: “cancer”, “chemotherapy” and “venous thromboembolism”. The complete search strategy is available in the Appendix A (Search Strategy). Studies were filtered to include those in the English language, conducted on humans, and restricted to disregard Phase I studies, accepting only those Phase II and above. Additional studies were also included through handsearching of references from included studies as well as from other sources such as Google Scholar and ResearchGate. We followed the Preferred Reporting Items for Systematic Reviews and Meta-Analyses (PRISMA) guidelines. This study was registered in Open Science, registration number is “yn5br” (https://osf.io/yn5br/ (accessed on 6 November 2022)). The PRISMA flowchart shows the studies included in the meta-analysis (Figure 1). PRISMA checklist is also provided in the Appendix A (PRISMA Checklist).

### 2.2. Inclusion and Exclusion Criteria

Studies were eligible for inclusion if they met the following criteria: (1) age ≥ 18 years; (2) patients with a confirmed diagnosis of cancer; (3) patients receiving chemotherapy; (4) patients not on prophylactic anticoagulation concomitant to chemotherapy, (5) availability of data on VTE occurrence noted in patients; and (6) studies with a sample size of >20 patients. The exclusion criteria were (1) studies not in English, (2) animal studies, (3) duplicated publications, (4) systematic reviews, meta-analyses, or narrative reviews; and (5) studies whereby relevant data on VTE occurrence not available. 

## 3. Data Extraction

Firstly, titles and abstracts were screened on EndNote™ (Clarivate, Philadelphia, PA, USA) to identify articles that were beyond the scope of this study, were systematic reviews of meta-analyses, or for other reasons failed to match the eligibility criteria before being excluded. Remaining articles were read in full-text and comprehensively assessed to determine eligibility for inclusion in this study, with screening conducted independently by two experienced investigators. In the case of disagreement between authors, a consensus was reached through discussion. A data extraction sheet was used to extract the following data from each study: (1) baseline demographics: author, year of publication, type of publication, country of lead author, study design, and study type; (2) study population: age of patients, sample size, baseline clinical characteristics, cancer phenotype, body location of cancer, cancer stage, and treatment agent, dose, duration, and frequency; (3) outcome measures: VTE occurrence. In the grading of VTE severity, most VTEs were classified as an adverse effect within a drug trial, and thus were graded via the Common Terminology Criteria for Adverse Events (CTCAE) scale of Grade 1 through 5. Although some studies were particular in grading each adverse event into individual categories of Grade 1/2/3/4/5, the majority of studies grouped Grades 1 and 2 together, and Grades 3/4/5 together. As such, in our meta-analysis, we have extracted data based on this later, more generalised method. Studies reporting on VTE in cancer, without prophylactic anticoagulation concomitant to chemotherapy, were included in the systematic review and meta-analysis.

### Quality Assessment of Included Studies

Using the modified Jadad analysis (MJA) criterion, the methodological quality of each study was assessed [25]. The MJA evaluates the quality of studies based upon: randomisation, blinding, description of withdrawals/dropouts, inclusion/exclusion criteria, assessment of adverse events and methods used for statistical analysis. Studies receive a score from 0–8 based upon their ability to fulfil aforementioned criteria [26]. The complete quality assessment of each study is available in the Appendix A (Jadad Analysis). Each study was also separately assessed for risk of funding bias using a 2-point scale that scored studies from 0 (low potential for bias) to 2 (high potential for bias) [27]. The absence of industry funding was not taken to signify an absence of bias, but the presence of industry funding or conflicts of interest was assumed to be an indicator of bias.

## 4. Statistical Analysis

Statistical analysis was performed using STATA (Version 13.0, StataCorp LLC, College Station, TX, USA). The purpose of this study was to determine the prevalence of VTE in cancer patients undergoing chemotherapy. As a result, the “metaprop” STATA command was utilised, pooling prevalence by performing a random-effects meta-analysis of proportions obtained from the individual studies [28]. The DerSimonian and Laird method was used for random effects modelling. In presenting the overall effects, forest plots were generated. Heterogeneity across the studies was estimated from the inverse-variance fixed-effect model and quantified using the I^2^ measure (I^2^ < 40% = low, 30–60% = moderate, 50–90% = substantial, and 75–100% = considerable). An overall meta-analysis was performed stratified by cancer phenotype to estimate the pooled prevalence of VTE in cancer patients undergoing chemotherapy. Besides, meta-analysis for individual cancer phenotypes were also performed provided there were minimum of 4 studies. The estimate of between-study variance (tau-squared or τ^2^) was also reported. Significance tests in the form of z-statistics and *p*-values were also reported. *p*-values less than 0.05 were considered significant.

## 5. Results

A total of 2643 and 85 studies were identified from PubMed and other sources, respectively. On screening of 2723 titles/abstracts, 2172 studies were excluded. Out of these, full texts of 551 studies were assessed for eligibility. Overall, 102 studies were included in the final synthesis.

### 5.1. Description of Included Studies

This meta-analysis included 102 studies, which reported on VTE prevalence within cancer patients undergoing chemotherapy, with a cumulative cohort of 30,671 patients (1773 with VTE, 28,898 without). Within a cohort of 20,420 patients for which data on sex was reported, 53.11% were male (sex data on 10,251 patients were not reported). Age ranged from 18–93 years. The mean age within a cohort of 8159 patients on which age data was reported was 59.59 years (standard deviation (SD) 33.74). Twenty-two cancer phenotypes were identified including bladder, blood, brain, breast, cervical, colorectal, endometrial, gastric, germ cell, head and neck, liver, lung, lymph, mesothelial, mixed, neuroendocrine, oesophageal, ovarian, pancreatic, prostate, renal, and skin.

The clinical characteristics of all included studies are shown in Table 1. Table 2 and Table 3 further delineate treatment dose, duration, and frequency. Tabulated results for the estimated pooled prevalence and crude prevalence rates stratified by cancer phenotype are shown in Table 4. The MJA and funding bias analysis for each study can be found in Appendix A.

### 5.2. Overall Prevalence of VTE in Cancer Patients Undergoing Chemotherapy

One hundred and two included studies, encompassing 30,671 patients, reported on the prevalence of VTE in cancer patients receiving chemotherapy only. The meta-analysis revealed a pooled estimated prevalence of 6%, ranging from 6% to 7% (ES 6%; 95% CI 6–7%; z = 18.53; *p* < 0.001) (Table 4 and Figure 2). Notably, there was considerable heterogeneity between the included studies (I^2^ = 91.84%, *p* < 0.001). The estimate of between-study variance (τ^2^) was 0.04. The estimated pooled prevalence of VTE in cancer patients undergoing chemotherapy was higher than the crude prevalence rates of 5.78% observed in this study. The heterogeneity chi^2^ was 1237.89 (*p* < 0.001, d.f. = 101). Figure 3 provides findings of the overall meta-analysis on the pooled estimated prevalence stratified by cancer phenotype.

### 5.3. Prevalence of VTE in Cancer Patients Stratified by Cancer Phenotype

#### 5.3.1. Prevalence of VTE in Bladder Cancer Patients

Four included studies encompassing 2700 patients reported on the prevalence of VTE in bladder cancer patients undergoing chemotherapy [35,87,91,116]. The meta-analysis revealed a pooled estimated prevalence of 18%, ranging from 10% to 28% (ES 18%; 95% CI 10–28%; z = 6.53; *p* < 0.001) (Table 4 and Appendix A). Notably, there was considerable heterogeneity between the included studies (I^2^ = 95.85%, *p* < 0.001). The estimate of between-study variance (τ^2^) was 0.05. The estimated pooled prevalence of VTE in blood cancer patients (18%) was higher than the crude prevalence rate of 11.30% observed in this study. The heterogeneity chi^2^ was 72.22 (*p* < 0.001, d.f. = 3).

#### 5.3.2. Prevalence of VTE in Blood Cancer Patients

Three included studies, encompassing 934 patients, reported on the prevalence of VTE in blood cancer patients [85,107,117]. However, a meta-analysis could not be performed due to insufficient number of studies. The crude prevalence rate of VTE in blood cancer patients was 10.81% (Table 4).

#### 5.3.3. Prevalence of VTE in Brain Cancer Patients

Eight included studies encompassing 3177 patients reported on the prevalence of VTE in brain cancer patients undergoing chemotherapy [29,30,58,89,93,98,127,129]. The meta-analysis revealed a pooled estimated prevalence of 4%, ranging from 4 to 5% (ES 4%; 95% CI 4–5%; z = 17.72; *p* < 0.001) (Table 4 and Appendix A). Notably, there was low heterogeneity between the included studies (I^2^ = 0.00%, *p* = 0.56). The estimate of between-study variance (τ^2^) was 0.00. The estimated pooled prevalence of VTE in brain cancer patients (4%) was lower than the crude prevalence rate of 5.19% observed in this study. The heterogeneity chi^2^ was 72.22 (*p* = 0.56, d.f. = 3).

#### 5.3.4. Prevalence of VTE in Breast Cancer Patients

Eight included studies encompassing 3082 patients reported on the prevalence of VTE in breast cancer patients undergoing chemotherapy [36,37,46,69,76,103,104,112]. The meta-analysis revealed a pooled estimated prevalence of 1%, ranging from 0% to 3% (ES 1%; 95% CI 0–3%; z = 4.17; *p* < 0.001) (Table 4 and Appendix A). Notably, there was substantial heterogeneity between the included studies (I^2^ = 73.31%, *p* < 0.001). The estimate of between-study variance (τ^2^) was 0.01. The estimated pooled prevalence of VTE in breast cancer patients (1%) was lower than the crude prevalence rate of 1.88% observed in this study. The heterogeneity chi^2^ was 26.23 (*p* < 0.001, d.f. = 7).

#### 5.3.5. Prevalence of VTE in Cervical Cancer Patients

Two included studies, encompassing 716 patients, reported on the prevalence of VTE in cervical cancer patients [63,124]. However, a meta-analysis could not be performed due to insufficient number of studies. The crude prevalence rate of VTE in cervical cancer patients was 6.42% (Table 4).

#### 5.3.6. Prevalence of VTE in Colorectal Cancer Patients

Fifteen included studies encompassing 5891 patients reported on the prevalence of VTE in colorectal cancer patients undergoing chemotherapy [34,42,44,53,55,59,88,102,106,110,111,118,122,125,126]. The meta-analysis revealed a pooled estimated prevalence of 5%, ranging from 3 to 7% (ES 5%; 95% CI 3–7%; z = 8.16; *p* < 0.001) (Table 4 and Appendix A). Notably, there was substantial heterogeneity between the included studies (I^2^ = 85.28%, *p* < 0.001). The estimate of between-study variance (τ^2^) was 0.02. The estimated pooled prevalence of VTE in colorectal cancer patients (5%) was higher than the crude prevalence rate of 4.69% observed in this study. The heterogeneity chi^2^ was 95.10 (*p* < 0.001, d.f. = 14).

#### 5.3.7. Prevalence of VTE in Endometrial Cancer Patients

Three included studies, encompassing 173 patients, reported on the prevalence of VTE in endometrial cancer patients [31,54,74]. However, a meta-analysis could not be performed due to insufficient number of studies. The crude prevalence rate of VTE in endometrial cancer patients was 11.56% (Table 4).

#### 5.3.8. Prevalence of VTE in Gastric Cancer Patients

Seven included studies encompassing 4932 patients reported on the prevalence of VTE in gastric cancer patients undergoing chemotherapy [57,78,86,94,99,119,130]. The meta-analysis revealed a pooled estimated prevalence of 9%, ranging from 5% to 15% (ES 9%; 95% CI 5–15%; z = 5.89; *p* < 0.001) (Table 4 and Appendix A). Notably, there was considerable heterogeneity between the included studies (I^2^ = 95.94%, *p* < 0.001). The estimate of between-study variance (τ^2^) was 0.05. The estimated pooled prevalence of VTE in gastric cancer patients (9%) was higher than the crude prevalence rate of 6.55% observed in this study. The heterogeneity chi^2^ was 147.92 (*p* < 0.001, d.f. = 6).

#### 5.3.9. Prevalence of VTE in Germ Cell Cancer Patients

One included study, encompassing 193 patients, reported on the prevalence of VTE in endometrial cancer patients [65]. However, a meta-analysis could not be performed due to insufficient number of studies. The crude prevalence rate of VTE in germ cell cancer patients was 2.07% (Table 4).

#### 5.3.10. Prevalence of VTE in Head and Neck Cancer Patients

Two included studies, encompassing 158 patients, reported on the prevalence of VTE in head and neck cancer patients [38,62]. However, a meta-analysis could not be performed due to insufficient number of studies. The crude prevalence rate of VTE in head and neck cancer patients was 1.27% (Table 4).

#### 5.3.11. Prevalence of VTE in Liver Cancer Patients

Two included studies, encompassing 347 patients, reported on the prevalence of VTE in liver cancer patients [43,109]. However, a meta-analysis could not be performed due to insufficient number of studies. The crude prevalence rate of VTE in liver cancer patients was 5.19% (Table 4).

#### 5.3.12. Prevalence of VTE in Lung Cancer Patients

Sixteen included studies encompassing 3228 patients reported on the prevalence of VTE in lung cancer patients undergoing chemotherapy [47,60,61,64,68,71,77,83,84,90,92,101,114,120,123]. The meta-analysis revealed a pooled estimated prevalence of 5%, ranging from 2 to 9% (ES 5%; 95% CI 2–9%; z = 4.32; *p* < 0.001) (Table 4 and Appendix A). Notably, there was considerable heterogeneity between the included studies (I^2^ = 93.22%, *p* < 0.001). The estimate of between-study variance (τ^2^) was 0.08. The estimated pooled prevalence of VTE in lung cancer patients (5%) was higher than the crude prevalence rate of 3.97% observed in this study. The heterogeneity chi^2^ was 221.28 (*p* < 0.001, d.f. = 15).

#### 5.3.13. Prevalence of VTE in Lymph Cancer Patients

Six included studies encompassing 699 patients reported on the prevalence of VTE in lymph cancer patients undergoing chemotherapy [41,45,50,51,72,97]. The meta-analysis revealed a pooled estimated prevalence of 4%, ranging from 2% to 7% (ES 4%; 95% CI 2–7%; z = 4.69; *p* < 0.001) (Table 4 and Appendix A). Notably, there was substantial heterogeneity between the included studies (I^2^ = 54.86%, *p* = 0.05). The estimate of between-study variance (τ^2^) was 0.01. The estimated pooled prevalence of VTE in lymph cancer patients (4%) was higher than the crude prevalence rate of 3.58% observed in this study. The heterogeneity chi^2^ was 11.08 (*p* = 0.05, d.f. = 5).

#### 5.3.14. Prevalence of VTE in Mesothelial Cancer Patients

Five included studies encompassing 1286 patients reported on the prevalence of VTE in mesothelial cancer patients undergoing chemotherapy [39,49,80,113,128]. The meta-analysis revealed a pooled estimated prevalence of 6%, ranging from 3% to 11% (ES 6%; 95% CI 3–11%; z = 5.24; *p* < 0.001) (Table 4 and Appendix A). Notably, there was substantial heterogeneity between the included studies (I^2^ = 84.17%, *p* < 0.001). The estimate of between-study variance (τ^2^) was 0.02. The estimated pooled prevalence of VTE in mesothelial cancer patients (6%) was higher than the crude prevalence rate of 4.82% observed in this study. The heterogeneity chi^2^ was 25.27 (*p* < 0.001, d.f. = 4).

#### 5.3.15. Prevalence of VTE in Neuroendocrine Cancer Patients

One included study, encompassing 113 patients, reported on the prevalence of VTE in neuroendocrine cancer patients [56]. However, a meta-analysis could not be performed due to insufficient number of studies. The crude prevalence rate of VTE in neuroendocrine cancer patients was 6.19% (Table 4).

#### 5.3.16. Prevalence of VTE in Oesophageal Cancer Patients

Two included studies, encompassing 328 patients, reported on the prevalence of VTE in oesophageal cancer patients [52,67]. However, a meta-analysis could not be performed due to insufficient number of studies. The crude prevalence rate of VTE in oesophageal cancer patients was 9.76% (Table 4).

#### 5.3.17. Prevalence of VTE in Ovarian Cancer Patients 

Six included studies encompassing 718 patients reported on the prevalence of VTE in ovarian cancer patients undergoing chemotherapy [40,79,82,96,115,121]. The meta-analysis revealed a pooled estimated prevalence of 8%, ranging from 5% to 12% (ES 8%; 95% CI 5–12%; z = 7.47; *p* = 0.02) (Table 4 and Appendix A). Notably, there was substantial heterogeneity between the included studies (I^2^ = 61.47%, *p* = 0.02). The estimate of between-study variance (τ^2^) was 0.01. The estimated pooled prevalence of VTE in ovarian cancer patients (8%) was lower than the crude prevalence rate of 8.22% observed in this study. The heterogeneity chi^2^ was 12.98 (*p* = 0.02, d.f. = 5).

#### 5.3.18. Prevalence of VTE in Pancreatic Cancer Patients

Three included studies, encompassing 144 patients, reported on the prevalence of VTE in pancreatic cancer patients [32,33,81]. However, a meta-analysis could not be performed due to insufficient number of studies. The crude prevalence rate of VTE in pancreatic cancer patients was 28.47% (Table 4).

#### 5.3.19. Prevalence of VTE in Prostate Cancer Patients

Three included studies, encompassing 1233 patients, reported on the prevalence of VTE in prostate cancer patients [95,100,108]. However, a meta-analysis could not be performed due to insufficient number of studies. The crude prevalence rate of VTE in prostate cancer patients was 2.11% (Table 4).

#### 5.3.20. Prevalence of VTE in Renal Cancer Patients

Two included studies, encompassing 198 patients, reported on the prevalence of VTE in renal cancer patients [48,105]. However, a meta-analysis could not be performed due to insufficient number of studies. The crude prevalence rate of VTE in renal cancer patients was 11.11% (Table 4).

#### 5.3.21. Prevalence of VTE in Skin Cancer Patients

One included study, encompassing 93 patients, reported on the prevalence of VTE in skin cancer patients [75]. However, a meta-analysis could not be performed due to insufficient number of studies. The crude prevalence rate of VTE in skin cancer patients was 7.53% (Table 4).

## 6. Discussion

Our meta-analysis revealed an overall pooled estimated prevalence of VTEs in cancer patients undergoing chemotherapy, as well for various cancer phenotypes. Our findings indicate that the estimated pooled prevalence of VTEs in cancer patients undergoing chemotherapy is approximately 6%, ranging from 5% to 7%, which is higher than the crude prevalence rate of 5.78%. To the best of our knowledge, this is one of the first reports in which prevalence estimates of VTE have been conducted on a relatively large cohort of patients. Our findings also reveal phenotypic variability in VTE risk, indicating need for prophylactic management of VTE risk in cancer patients undergoing chemotherapy, with certain phenotypes of cancer such as bladder, gastric and ovarian posing particularly high risks of VTE.

One explanation for why cancer patients have a higher risk of having VTE is that tumours can express various procoagulant molecules and alter tissue factor expression [131,132]. Certain tumours may also raise the risk of thrombosis by compressing blood vessels, changing blood flow, or causing injury to the vascular endothelium through intravascular growth [9]. Subsequent cancer diagnosis within the first year of first VTE diagnosis have been reported in up to 10% of patients [133]. Therefore, VTE, especially in the lower limbs, can also be useful as marker for occult cancer [134].

This pooled estimate of 6% is higher than other estimates of 2.3% prevalence rates of VTE in cancer patients in the first 12 months after their diagnosis, with other estimates ranging from 4–20% of cancer patients developing VTE in their lifetime [12,135]. Amongst the normal population, VTE prevalence is at 1–2% [136]. In a retrospective study on 40,787,000 hospitalised cancer patients from 1979 through 1999, patients with malignancy were found to have a 2% prevalence of thromboembolism, although, were not necessarily on chemotherapy or radiotherapy treatment [137]. This suggests that cancer itself, without the interference of external treatment regimens, may not pose a significant risk to VTE but rather, it is the accompanying therapies which may confer additional VTE risk.

The prevalence of VTE may vary across cancer phenotype. This is of clinical interest as it may aid in the risk-staging and appropriate tailored management specific to cancer phenotype. We found that the pooled VTE prevalence varied across cancer phenotype in the range of 1–18%, with lowest prevalence of 1% observed in breast and head and neck cancer and highest prevalence of 18% observed in bladder cancer. This indicates a need for more aggressive VTE screening for specific cancer phenotypes. From a policy standpoint, beyond the hospital-based risk factors, such as recent surgery, cancer, and congestive heart failure, to prevent VTE, dietary counselling as well as public health strategies around encouraging the adoption of heart-healthy habits for cancer patients undergoing chemotherapy may be beneficial [138]. Moreover, concomitant preventative measures targeting arterial thrombosis and VTE are also important [2].

### 6.1. Pathophysiology of VTE in Cancer Patients

The pathophysiological process behind VTE prevalence in cancer patients is multifaceted and can be attributed to multiple aetiological pathways, spanning the hypercoagulable state induced by malignancy itself to the thrombotic risk posed by treatment regimens of chemotherapy and radiotherapy [139]. The inflammatory state induced by malignancy, stemming from tumour biology and activation of the coagulation cascade, increases cancer patients’ risk of VTE occurrence [139]. On a molecular level, several factors increase the risk of VTE, with increased concentrations of procoagulants on a cellular level amplifying thrombosis prevalence. These include tissue factor, microparticles, plasminogen activator inhibitor-1, cancer procoagulant, mucin, tumour-derived platelet agonists and inflammatory cytokines such as IL-6, IL-8 and IL-10 [140,141,142,143,144,145]. Alterations to thrombomodulin expression due to interference from tumour necrosis factor-a and IL-1B also contribute to a prothrombotic state [146].

### 6.2. Chemotherapy and VTE

Chemotherapy has been shown to increase VTE risk by six-fold in cancer patients [12]. Multiple chemotherapy drugs which are used to treat cancer are associated with increased thrombotic events [147]. Cisplatin is a major component of several treatment regimens—and its thrombotic potential and vascular toxicity has been identified since 1986 [148]. Through direct drug-induced damage to the endothelium and by indirectly increasing the expression of TF procoagulant activity of monocytes and macrophages, chemotherapy poses a serious risk of increasing VTE within a cancer patient [148].

## 7. Limitations

There are several limitations to this study due to variance across the quality of studies included and therefore ability to accurately process the data extracted. Firstly, the types of studies included vary from being retrospective in design to being randomised controlled trials (RCT)—therefore, whilst some studies noted VTE as one of multiple adverse effects within an RCT for a novel chemotherapy regimen, others purely sought to document VTE occurrence within a cohort of cancer patients which oftentimes varied in cancer phenotype, staging and treatment. Furthermore, whilst some studies were robust in being double-blinded, randomised, placebo-controlled and multi-institutional, others were single institution studies conducted on a relatively small cohort size, without an appropriate control group or blinding. As such, this wide variance in included study quality could confound the overall pooled estimated prevalence. Within these studies, their documentation of patients’ cancer history is highly limited. Reporting of time of diagnosis to treatment, the duration, drug regimen and frequency of previous treatments are inconsistent and rarely available. As such, it is difficult to determine whether previous treatment regimens played a confounding role in patients developing VTE. We are also unable to determine whether variance in time of diagnosis to treatment plays a role. Moreover, there was a lack of standardised reporting and insufficient detail in the description of VTEs across the studies. As the majority of studies included were RCTs, VTEs were often a side effect as opposed to the focus of the study, and thus less attention was given toward the VTE. VTE pooled prevalence stratified by cancer severity grade was not investigated in this meta-analysis. In few cases, studies neglected to document the severity of thrombotic events altogether—in which we have assumed a Grade 3/4/5 event in that case. Besides, detailed analysis into the association between severity of VTE event, and any relationship with cancer phenotype, time to treatment, staging, drug regimen, or patient profile could not be performed. Additionally, documentation on certain groups such as atrial fibrillation and VTE recurrence were not available across all studies. It would be ideal to understand whether VTE events occurred before therapy, during, or how long post-diagnosis and post-treatment. As such, more robust future studies with more detailed information and reporting on VTE occurrence, recurrence and adverse effects is necessary. Another limitation of this study and in the studies gathered is the lack of accounting for baseline underlying comorbidities in all the patients. Important factors such as atherosclerosis, cardiovascular disease, histories of smoking, histories of VTE, obesity and age were not detailed in the original studies.

Although patients on prophylactic anticoagulation concomitant to chemotherapy were not included in this study, we acknowledge that previous history of anticoagulation may presumably not have been reported in some studies. In light of recent guidelines [19], as adherence to prophylactic anticoagulation grows to reduce VTE risk, it is likely that VTE prevalence will show a downward trajectory. Finally, for the patients who did experience VTE prevalence, often these patients were not followed longitudinally for VTE recurrence, and specific time to disease progression and overall survival. Typically, follow-up was not provided for longer-term complications and recurrence. The discrepancy in protocol for VTE diagnosis and follow-up between hospitals and studies leads to the inconsistent reporting and treatment of patients across the clinical decision making, imaging and diagnosis pipeline. Despite these limitations, the use of random-effects modelling would have mitigated some of the random biases and risks above.

## 8. Conclusions

In conclusion, this meta-analysis demonstrated a pooled prevalence estimate of 6%, with a range of 5% to 7%, of VTEs amongst cancer patients undergoing chemotherapy. Our study indicates there is substantial risk of developing VTE as a cancer patient on chemotherapy showing a compelling need for robust screening and subsequent prophylactic management to prevent future VTE. More efforts should be undertaken to implement adherence of American Society of Haematology guidelines on VTE risks and management in cancer patients undergoing chemotherapy [19].

## Figures and Tables

**Figure 1 diagnostics-12-02954-f001:**
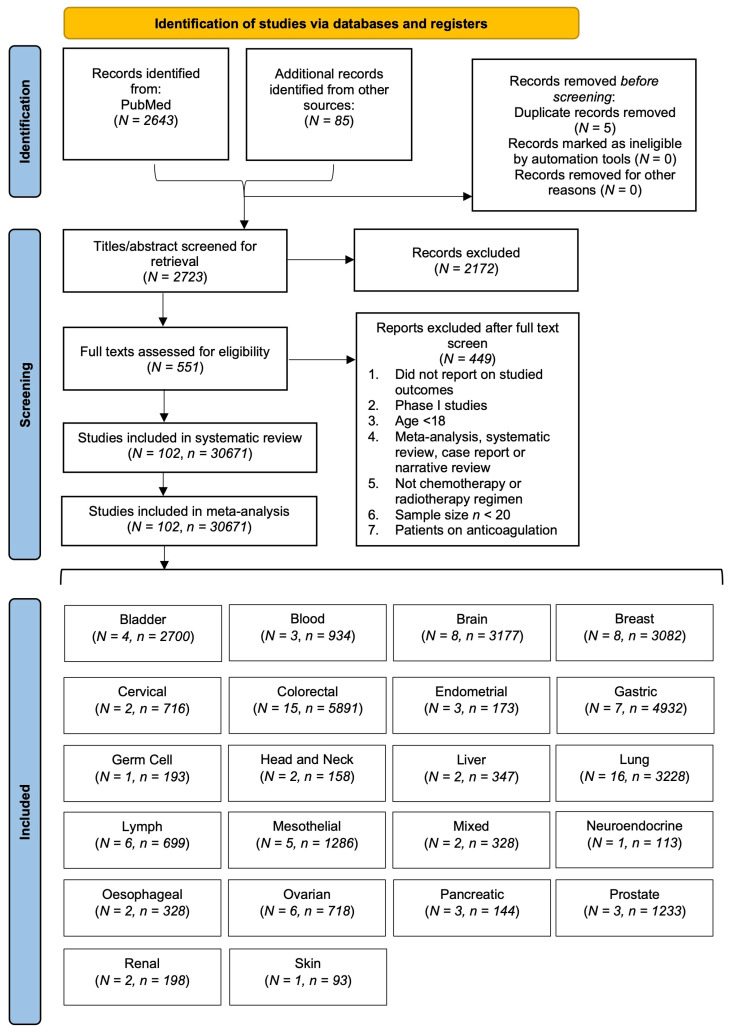
The PRISMA flowchart showing the steps followed during the study selection process. Abbreviations: *N*: number of studies; *n*: number of patients.

**Figure 2 diagnostics-12-02954-f002:**
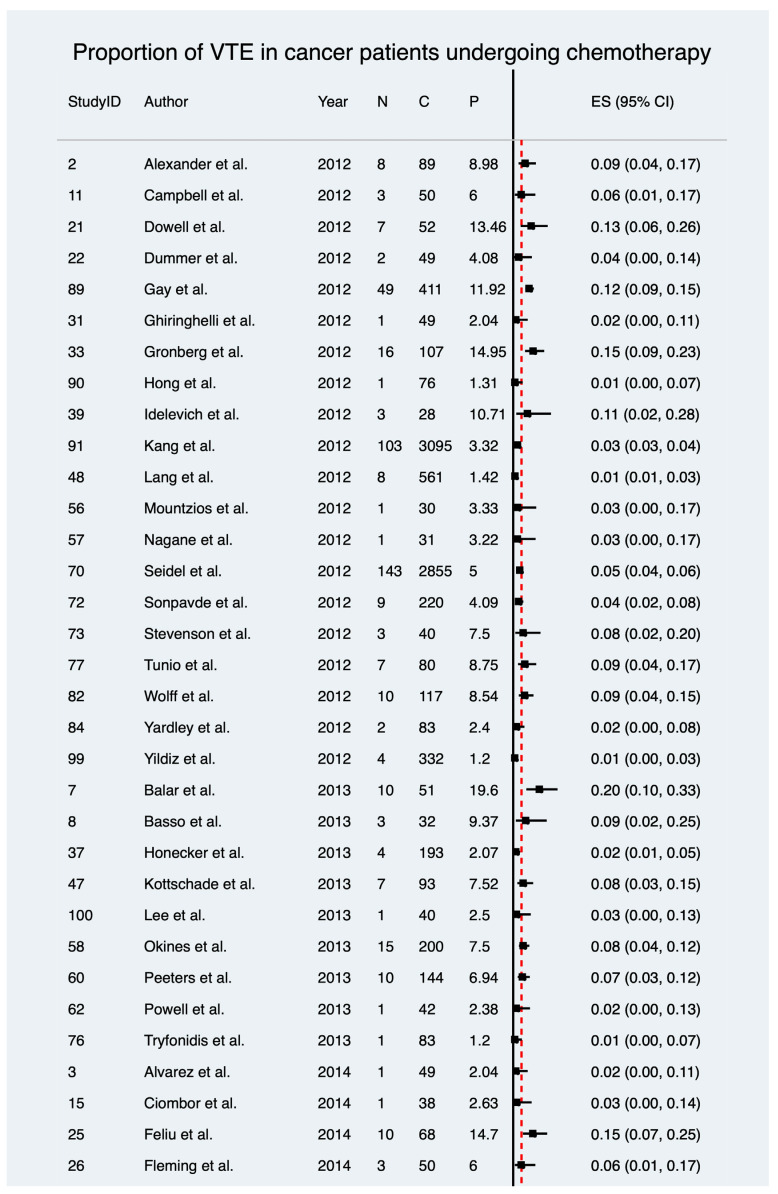
Forest plot showing the estimated overall pooled prevalence of VTE in cancer patients undergoing chemotherapy [29,30,31,32,33,34,35,36,37,38,39,40,41,42,43,44,45,46,47,48,49,50,51,52,53,54,55,56,57,58,59,60,61,62,63,64,65,66,67,68,69,70,71,72,73,74,75,76,77,78,79,80,81,82,83,84,85,86,87,88,89,90,91,92,93,94,95,96,97,98,99,100,101,102,103,104,105,106,107,108,109,110,111,112,113,114,115,116,117,118,119,120,121,122,123,124,125,126,127,128]. Abbreviations: VTE: venous thromboembolism.

**Figure 3 diagnostics-12-02954-f003:**
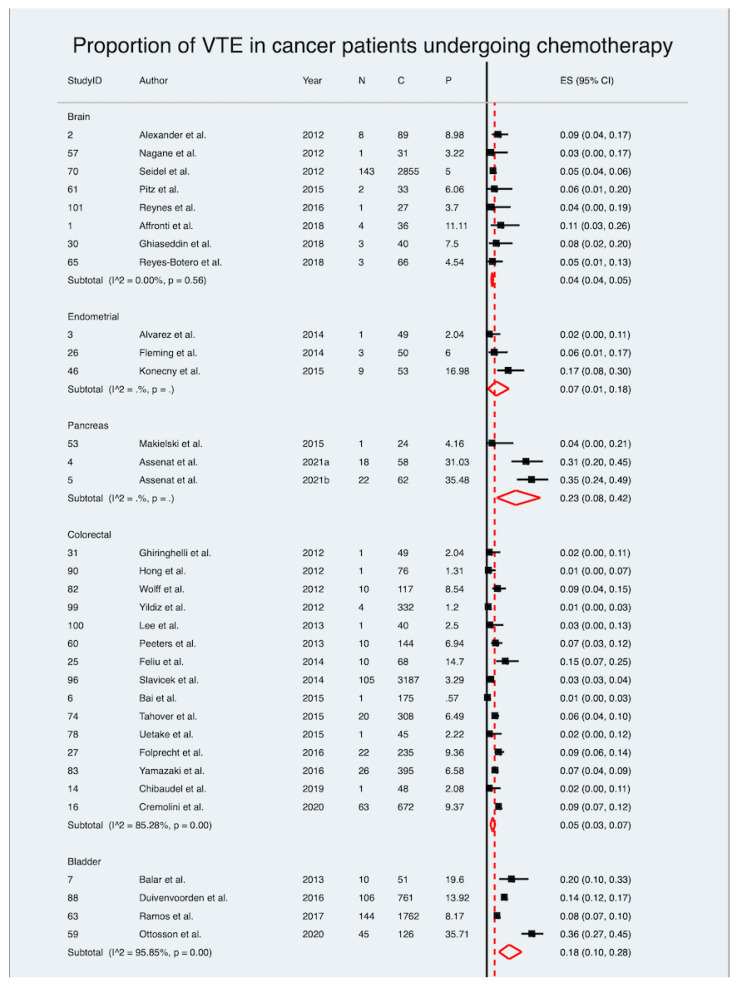
Forest plot showing the estimated pooled prevalence of VTE in cancer patients stratified by cancer phenotype [29,30,31,32,33,34,35,36,37,38,39,40,41,42,43,44,45,46,47,48,49,50,51,52,53,54,55,56,57,58,59,60,61,62,63,64,65,66,67,68,69,70,71,72,73,74,75,76,77,78,79,80,81,82,83,84,85,86,87,88,89,90,91,92,93,94,95,96,97,98,99,100,101,102,103,104,105,106,107,108,109,110,111,112,113,114,115,116,117,118,119,120,121,122,123,124,125,126,127,128]. Abbreviations: VTE: venous thromboembolism.

**Table 1 diagnostics-12-02954-t001:** Clinical characteristics of studies included in the meta-analysis.

Study ID	Author	Year	Study Design	Study Phase	Country	Cancer Phenotype	Cancer Phenotype, Body	Age (Median)	Age (Range)	Number of Males (%)	C	N	P
1	Affronti et al. [29]	2018	Prospective, single centre	II	USA	Recurrent grade IV malignant glioma	Brain	55.5	27–74	61.11	36	4	11.11
2	Alexander et al. [30]	2012	Prospective, single centre	II	USA	Newly diagnosed glioblastoma	Brain	N/A	N/A	62	89	8	8.99
3	Alvarez et al. [31]	2014	Prospective, single centre	II	USA	Recurrent or persistent endometrial carcinoma	Endometrial	63	35–80	0	49	1	2.04
4	Assenat et al. [32]	2021a	Prospective, multicentre	II	USA	Metastatic pancreatic cancer	Pancreas	60	34–72	50	58	18	31.03
5	Assenat et al. [33]	2021b	Prospective, multicentre	II	France	Metastatic pancreatic cancer	Pancreas	62	35–77	59.7	62	22	35.48
6	Bai et al. [34]	2015	Prospective, single centre	Unspecified	China	Metastatic colorectal cancer	Colorectal	55	20–79	63.4	175	1	0.57
7	Balar et al. [35]	2013	Prospective, single centre	II	USA	Advanced unresectable/metastatic urothelial cancer	Bladder	67	42–83	72.5	51	10	19.61
8	Basso et al. [36]	2013	Prospective, multicentre	Unspecified	Italy	Locally advanced/metastatic breast cancer	Breast	78	70–93	0	32	3	9.38
9	Bear et al. [37]	2015	Prospective, multicentre	III	USA	Early HER2-negative breast cancer	Breast	N/A	N/A	0	1206	37	3.07
10	Buxo et al. [38]	2018	Retrospective, single centre	Unspecified	Spain	Recurrent or metastatic head and neck squamous cell carcinoma	Head and Neck	N/A	N/A	N/A	104	1	0.96
11	Campbell et al. [39]	2012	Prospective, multicentre	II	USA	Malignant mesothelioma	Mesothelium	N/A	N/A	84	50	3	6.00
12	Chekerov et al. [40]	2018	Prospective, multicentre	II	Germany	Platinum-resistant ovarian cancer	Ovary	N/A	N/A	0	174	10	5.75
13	Chen et al. [41]	2015	Prospective, single centre	II	USA	Relapsed/refractory indolent non-Hodgkin lymphoma	Lymph	62	44–85	54	28	4	14.29
14	Chibaudel et al. [42]	2019	Prospective, multicentre	II	France	Metastatic colorectal cancer	Colorectal	62.9	32–86	53.1	48	1	2.08
15	Ciombor et al. [43]	2014	Prospective, multicentre	II	USA	Hepatocellular carcinoma	Liver	59	23–76.5	71.1	38	1	2.63
16	Cremolini et al. [44]	2020	Prospective, multicentre	III	Italy	Unresectable metastatic colorectal cancer	Colorectal	N/A	N/A	N/A	672	63	9.38
17	de Vos et al. [45]	2014	Prospective, multicentre	II	USA	Diffuse large B-cell lymphoma	Lymph	72	18–85	61	46	3	6.52
18	DeCensi et al. [46]	2019	Prospective, multicentre	III	Italy	Ductal carcinoma in situ	Breast	N/A	N/A	0	500	2	0.40
19	Deschenes-Simard et al. [47]	2021	Retrospective, multicentre	Unspecified	Canada	Non-small-cell lung cancer	Lung	66.7	N/A	54.3	593	64	10.79
20	Donskov et al. [48]	2018	Prospective, multicentre	IIIb	Denmark	Metastatic renal cell carcinoma	Renal	N/A	N/A	N/A	118	15	12.71
21	Dowell et al. [49]	2012	Prospective, multicentre	II	USA	Advanced malignant mesothelioma	Mesothelium	66	24–81	85	52	7	13.46
22	Dummer et al. [50]	2012	Prospective, multicentre	II	Switzerland	Primary cutaneous T-cell lymphoma, mycosis fungoides	Lymph	N/A	N/A	N/A	49	2	4.08
23	Duvic et al. [51]	2015	Prospective, single centre	II	USA	Cutaneous T-cell lymphoma and lymphomatoid papulosis	Lymph	59.5	31–77	54	48	2	4.17
24	Fehr et al. [52]	2020	Prospective, multicentre	III	Switzerland	Locally advanced oesophageal cancer	Oesophageal	61	36–75	88	300	29	9.67
25	Feliu et al. [53]	2014	Prospective, multicentre	II	Spain	Metastatic colorectal cancer	Colorectal	75.6	70.5–85.4	65	68	10	14.71
26	Fleming et al. [54]	2014	Prospective, multicentre	II	USA	Endometrial cancer	Endometrial	N/A	N/A	0	71	10	14.08
27	Folprecht et al. [55]	2016	Prospective, multicentre	II	Germany	Metastatic colorectal cancer	Colorectal	62.5	29–87	61	235	22	9.36
28	Frizziero et al. [56]	2019	Retrospective, multicentre	Unspecified	UK	Poorly differentiated neuroendocrine carcinomas	Neuroendocrine	65.8	24–88	63.7	113	7	6.19
29	Fuchs et al. [57]	2019	Prospective, multicentre	III	USA	Metastatic, HER2-negative gastric or gastrooesophageal junction adenocarcinoma	Gastric	N/A	N/A	N/A	645	99	15.35
30	Ghiaseddin et al. [58]	2018	Prospective, single centre	II	USA	Recurrent, grade 4 malignant glioma	Brain	52.4	32–74	60	40	3	7.50
31	Ghiringhelli et al. [59]	2012	Prospective, single centre	II	France	Metastatic colorectal cancer	Colorectal	63	25–79	53	49	1	2.04
32	Goss et al. [60]	2016	Prospective, multicentre	II	Canada	EGFR Thr790Met-positive advanced non-small-cell lung cancer	Lung	64	35–88	31	210	1	0.48
33	Gronberg et al. [61]	2012	Prospective, multicentre	II	Norway	Brain metastases from lung cancer	Lung	N/A	N/A	N/A	107	16	14.95
34	Guigay et al. [62]	2015	Prospective, multicentre	II	France	Recurrent or metastatic head and neck squamous cell carcincoma	Head and Neck	N/A	N/A	96.3	54	1	1.85
35	He et al. [63]	2020	Retrospective, single centre	Unspecified	China	Advanced cervical cancer	Cervix	N/A	N/A	0	264	24	9.09
36	Hirsch et al. [64]	2017	Prospective, multicentre	II	USA	Advanced squamous cell non-small-cell lung cancer	Lung	N/A	N/A	N/A	109	3	2.75
37	Honecker et al. [65]	2013	Retrospective, multicentre	Unspecified	Germany	Germ cell tumour	Germ Cell	35	18–83	N/A	193	4	2.07
38	Hu et al. [66]	2015	Prospective, single centre	II	China	Advanced non-small-cell lung cancer	Lung	59.6	32–83	55.4	56	12	21.43
39	Idelevich et al. [67]	2012	Prospective, single centre	II	Israel	Locally advanced resectable esophageal cancer	Oesophageal	N/A	N/A	82	28	3	10.71
40	Ikemura et al. [68]	2015	Prospective, single centre	II	Japan	Advanced non-small-cell lung cancer	Lung	59.5	35–74	80.6	31	1	3.23
41	Ishida et al. [69]	2015	Prospective, multicentre	Unspecified	Japan	Metastatic breast cancer	Breast	62	41–85	0	117	1	0.85
42	Kakkos et al. [70]	2020	Prospective, multicentre	Unspecified	Greece	Various—lung, pancreatic, ovarian, prostate	Mixed	N/A	N/A	N/A	231	17	7.36
43	Karavasilis et al. [71]	2014	Prospective, multicentre	II	Greece	Metastatic non-small-cell lung cancer	Lung	64	N/A	N/A	50	1	2.00
44	Kim et al. [72]	2018	Prospective, multicentre	Unspecified	USA	Previously treated cutaneous T-cell lymphoma	Lymph	N/A	N/A	N/A	372	7	1.88
45	Kitayama et al. [73]	2017	Prospective, single centre	Unspecified	Japan	Mixed	Mixed	65	N/A	48.5	97	29	29.90
46	Konecny et al. [74]	2015	Prospective, multicentre	II	USA	Metastatic endometrial cancer	Endometrial	N/A	N/A	0	53	9	16.98
47	Kottschade et al. [75]	2013	Prospective, multicentre	II	USA	Unresectable metastatic melanoma	Skin	N/A	N/A	N/A	93	7	7.53
48	Lang et al. [76]	2012	Prospective, multicentre	III	Hungary	Locally recurrent/metastatic breast cancer	Breast	N/A	N/A	0	561	8	1.43
49	Lara et al. [77]	2016	Prospective, multicentre	II	USA	Advanced non-small-cell lung cancer	Lung	N/A	N/A	N/A	59	1	1.69
50	Larsen et al. [78]	2015	Prospective, single centre	Unspecified	Denmark	Gastric, esophageal, gastro-oesophageal	Gastric	64	35–84	75.2	129	21	16.28
51	Lee et al. [79]	2020	Prospective, multicentre	II	USA	Recurrent ovarian cancer	Ovary	N/A	27–79	0	54	5	9.26
52	Maio et al. [80]	2017	Prospective, multicentre	IIb	Italy	Relapsed malignant mesothelioma	Mesothelium	66	60–72	74	571	17	2.98
53	Makielski et al. [81]	2015	Prospective, multicentre	II	USA	Advanced pancreatic cancer	Pancreas	63	48–83	N/A	24	1	4.17
54	Matsumoto et al. [82]	2015	Prospective, multicentre	II	Japan	Platinum-resistant taxane-pretreated ovarian cancer	Ovary	58	31–75	0	60	1	1.67
55	Michelsen & Sorensen [83]	2015	Prospective, single centre	Unspecified	Denmark	Advanced non-small-cell lung cancer	Lung	N/A	N/A	N/A	42	10	23.81
56	Mountzios et al. [84]	2012	Prospective, multicentre	II	Greece	Chemoresistant relapsed small cell lung cancer	Lung	64	43–82	90	30	1	3.33
57	Nagane et al. [85]	2012	Prospective, single centre	II	Japan	Recurrent malignant glioma	Brain	54	23–72	51.6	31	1	3.23
58	Okines et al. [86]	2013	Prospective, multicentre	II/III	UK	Localised gastro-oesophageal adenocarcinoma	Gastric	64	40–80	82	200	15	7.50
59	Ottosson et al. [87]	2020	Prospective, multicentre	Unspecified	Sweden	Muscle-invasive urinary bladder cancer	Bladder	N/A	N/A	80.6	126	45	35.71
60	Peeters et al. [88]	2013	Prospective, multicentre	II	Belgium	Metastatic colorectal cancer	Colorectal	N/A	N/A	N/A	144	10	6.94
61	Pitz et al. [89]	2015	Prospective, multicentre	II	Canada	Glioblastoma	Brain	56	35–78	63.6	33	2	6.06
62	Powell et al. [90]	2013	Prospective, single centre	II	USA	Advanced, refractory non-small-cell lung cancer	Lung	62.5	36–80	42.9	42	1	2.38
63	Ramos et al. [91]	2017	Retrospective, multicentre	Unspecified	USA	Metastatic urothelial carcinoma	Bladder	N/A	N/A	77.5	1762	144	8.17
64	Reck et al. [92]	2014	Prospective, multicentre	III	Germany	Non-small-cell lung cancer	Lung	N/A	N/A	N/A	1314	3	0.23
65	Reyes-Botero et al. [93]	2018	Prospective, multicentre	II	France	Newly diagnosed glioblastoma	Brain	76	70–87	36	66	3	4.55
66	Rivera et al. [94]	2015	Prospective, multicentre	II	Spain	Advanced gastric cancer	Gastric	73.3	40–87	74.41860465	43	4	9.30
67	Saad et al. [95]	2021	Prospective, multicentre	III	Canada	Metastatic, castration-resistant prostate cancer	Prostate	N/A	N/A	100	982	15	1.53
68	Salinaro et al. [96]	2020	Prospective, multicentre	Unspecified	USA	Advanced epithelial ovarian cancer	Ovary	64.8	34–84	0	230	16	6.96
69	Salles et al. [97]	2020	Prospective, multicentre	II	France	Relapsed or refractory diffuse large B-cell lymphoma	Lymph	72	62–76	54	156	7	4.49
70	Seidel et al. [98]	2012	Prospective, multicentre	Unspecified	Germany	Glioma	Brain	N/A	N/A	N/A	2855	143	5.01
71	Slagter et al. [99,100]	2020	Prospective, multicentre	Unspecified	Netherlands	Gastric cancer	Gastric	N/A	N/A	N/A	781	78	9.99
72	Sonpavde et al. [100]	2012	Prospective, multicentre	II	USA	Metastatic castration-resistant prostate cancer	Prostate	N/A	N/A	100	220	9	4.09
73	Stevenson et al. [101]	2012	Prospective, single centre	II	USA	Advanced, non-squamous non-small-cell lung cancer	Lung	65.3	35–80	46	40	3	7.50
74	Tahover et al. [102]	2015	Prospective, single centre	Unspecified	Israel	Metastatic colorectal cancer	Colorectal	N/A	N/A	N/A	308	20	6.49
75	Tan et al. [103]	2021	Prospective, multicentre	III	USA	HER2-positive early breast cancer	Breast	N/A	N/A	0	500	4	0.80
76	Tryfonidis et al. [104]	2013	Prospective, multicentre	II	Greece	Metastatic breast cancer HER-2 negative	Breast	62	23–75	0	83	1	1.20
77	Tunio et al. [105]	2012	Prospective, single centre	II	Pakistan	Metastatic renal cell carcinoma	Renal	51.11	23–73	73.8	80	7	8.75
78	Uetake et al. [106]	2015	Prospective, multicentre	II	Japan	Metastatic colorectal cancer	Colorectal	62.5	39–80	58.7	45	1	2.22
79	Usmani et al. [107]	2019	Prospective, multicentre	III	USA	Multiple myeloma	Blood	N/A	N/A	N/A	301	2	0.66
80	Vaishampayan et al. [108]	2014	Prospective, single centre	II	USA	Metastatic castrate-resistant prostate cancer	Prostate	67	50–85	100	31	2	6.45
81	Valle et al. [109]	2021	Prospective, multicentre	II	UK	Locally advanced or metastatic biliary tract cancer	Liver	N/A	N/A	N/A	309	17	5.50
82	Wolff et al. [110]	2012	Prospective, multicentre	II	USA	Metastatic colorectal cancer	Colorectal	N/A	N/A	N/A	117	10	8.55
83	Yamazaki et al. [111]	2016	Prospective, multicentre	III	Japan	Metastatic colorectal cancer	Colorectal	N/A	N/A	N/A	395	26	6.58
84	Yardley et al. [112]	2012	Prospective, multicentre	II	USA	Advanced breast cancer	Breast	N/A	35–83	0	83	2	2.41
85	Zalcman et al. [113]	2016	Prospective, multicentre	III	France	Newly diagnosed pleural mesothelioma	Mesothelium	N/A	N/A	N/A	448	15	3.35
86	Baggstrom et al. [114]	2017	Prospective, multicentre	III	USA	Non-small cell lung cancer	Lung	66	25–89	56	210	1	0.48
87	Chavan et al. [115]	2017	Retrospective, single-centre	Unspecified	China	Epithelial ovarian cancer	Ovary	N/A	26–75	0	144	20	13.89
88	Duivenvoorden et al. [116]	2016	Retrospective, multicentre	Unspecified	USA	Muscle invasive bladder cancer	Bladder	N/A	N/A	74.8	761	106	13.93
89	Gay et al. [117]	2010	Retrospective, multicentre	Unspecified	USA	Newly diagnosed multiple myeloma	Blood	N/A	N/A	N/A	411	49	11.92
90	Hong et al. [118]	2012	Prospective, multicentre	II	South Korea	Metastatic colorectal cancer	Colorectal	57	31–75	51.3	76	1	1.32
91	Kang et al. [118]	2012	Retrospective, single-centre	Unspecified	South Korea	Advanced gastric cancer	Gastric	57	18–88	66	3095	103	3.33
92	Li et al. [119]	2017	Prospective, multicentre	II	USA	Metastatic gastroesophageal adenocarcinoma	Gastric	62	27–79	79%	39	3	7.69
93	Martella et al. [85]	2022	Retrospective, multicentre	Unspecified	Italy	Newly diagnosed adult acute myeloid leukaemia	Blood	N/A	N/A	52	222	50	22.52
94	Matikas et al. [120]	2016	Prospective, multicentre	IV	Greece	Advanced non-small cell lung cancer	Lung	63	38–84	74.8	314	9	2.87
95	Monk et al. [121]	2018	Prospective, multicentre	II	USA	Recurrent or persistent platinum-resistant ovarian, fallopian tube or primary peritoneal cancer	Ovary	N/A	N/A	0	56	7	12.50
96	Slavicek et al. [122]	2014	Retrospective, multicentre	Unspecified	Czech Republic	Metastatic colorectal cancer	Colorectal	N/A	N/A	62.6	3187	105	3.29
97	Tachihara et al. [123]	2020	Prospective, multicentre	II	Japan	Resected nonsquamous non-small celll lung cancer	Lung	66	57–75	57.1	21	1	4.76
98	Tewari et al. [124]	2018	Prospective, multicentre	III	USA	Advanced cervical cancer	Cervix	N/A	N/A	0	452	22	4.87
99	Yildiz et al. [125]	2012	Retrospective, multicentre	Unspecified	Turkey	Metastatic colorectal cancer	Colorectal	53	18–74	61.7	332	4	1.20
100	Lee et al. [126]	2013	Prospective, multicentre	Unspecified	Taiwan	Metastatic colorectal cancer	Colorectal	57	32–87	62.5	40	1	2.50
101	Reynes et al. [127]	2016	Prospective, multicentre	II	UK	Recurrent glioblastoma	Brain	56	42–77	70.4	27	1	3.70
102	Pinto et al. [128]	2021	Prospective, multicentre	II	Italy	Malignant pleural mesothelioma	Mesothelium	69	44–81	74	165	20	12.12

Abbreviations: C: total number of patients, N: number of venous thromboembolism cases, P = crude prevalence of venous thromboembolism in individual studies, GI: gastrointestinal, VTE: venous thromboembolism, II: phase-two study, III: phase-three study.

**Table 2 diagnostics-12-02954-t002:** Prevalence of venous thromboembolism stratified by treatment regimen.

Study ID	Author	Year	Cancer Phenotype, Body	Treatment Agent	C	N	P	Grade 1/2 (n)	Grade 3/4/5 (n)
1	Affronti et al. [29]	2018	Brain	Bevacizumab with rilotumumab	36	4	11.11	0	4
2	Alexander et al. [30]	2012	Brain	Thalidomide	89	8	8.99	0	8
3	Alvarez et al. [31]	2014	Endometrial	Bevacizumab + temsirolimus	49	1	2.04	0	1
4	Assenat et al. [32]	2021a	Pancreas	Nab-paclitaxel/gemcitabine and FOLFIRINOX	58	18	31.03	8	10
5	Assenat et al. [33]	2021b	Pancreas	Gemcitabine, trastuzumab plus erlotinib	62	22	35.48	0	22
6	Bai et al. [34]	2015	Colorectal	mFOLFOX-6 or XELOX or FOLFIRI with bevacizumab	175	1	0.57	0	1
7	Balar et al. [35]	2013	Bladder	Gemcitabine, carboplatin and bevacizumab	51	10	19.61	0	10
8	Basso et al. [36]	2013	Breast	Liposomal doxorubicin	32	3	9.38	2	1
9	Bear et al. [37]	2015	Breast	Various	1206	37	3.07	0	37
10	Buxo et al. [38]	2018	Head and Neck	Carboplatin, cetuximab and tegafur	104	1	0.96	0	1
11	Campbell et al. [39]	2012	Mesothelium	Cediranib	50	3	6.00	0	3
12	Chekerov et al. [40]	2018	Ovary	Sorafenib plus topotecan versus placebo plus topotecan	174	10	5.75	5	5
13	Chen et al. [41]	2015	Lymph	Vorinostat and rituximab	28	4	14.29	0	4
14	Chibaudel et al. [42]	2019	Colorectal	Aflibercept with FOLFOX (folinic acid, fluorouracil, oxaliplatin) followed by maintenance with fluoropyrimidine	48	1	2.08	0	1
15	Ciombor et al. [43]	2014	Liver	Bortezomib plus doxorubicin	38	1	2.63	0	1
16	Cremolini et al. [44]	2020	Colorectal	mFOLFOX6 and bevacizumab followed by FOLFIRI plus bevacizumab after disease progression, or FOLFOXIRI and bevacizumab, followed by the same regimen after disease progression	672	63	9.38	32	31
17	de Vos et al. [45]	2014	Lymph	Dacetuzumab	46	3	6.52	0	3
18	DeCensi et al. [46]	2019	Breast	Tamoxifen	500	2	0.40	0	2
19	Deschenes-Simard et al. [47]	2021	Lung	Various immune checkpoint inhibitors including nivolumab, pembrolizumab, atezolizumab, avelumab, durvalumab, ipilimumab, tremelimumab, and M7824.	593	64	10.79	0	64
20	Donskov et al. [48]	2018	Renal	Interleukin-2 and interferon-a with or without bevacizumab	118	15	12.71	0	15
21	Dowell et al. [49]	2012	Mesothelium	Cisplatin, pemetrexed and bevacizumab	52	7	13.46	0	7
22	Dummer et al. [50]	2012	Lymph	Pegylated liposomal doxorubicin	49	2	4.08	0	2
23	Duvic et al. [51]	2015	Lymph	Brentuximab Vedotin	48	2	4.17	0	2
24	Fehr et al. [52]	2020	Oesophageal	Docetaxel and cisplatin,	300	29	9.67	13	16
25	Feliu et al. [53]	2014	Colorectal	Bevacizumab, oxaliplatin and oral capecitabine	68	10	14.71	3	7
26	Fleming et al. [54]	2014	Endometrial	Temsirolimus plus megestrol acetate/tamoxifen	71	10	14.08	0	10
27	Folprecht et al. [55]	2016	Colorectal	mFOLFOX6 with or without aflibercept	235	22	9.36	1	21
28	Frizziero et al. [56]	2019	Neuroendocrine	Carboplatin and etoposide	113	7	6.19	0	7
29	Fuchs et al. [57]	2019	Gastric	Cisplatin and capecitabine, and either ramucirumab or placebo	645	99	15.35	0	99
30	Ghiaseddin et al. [58]	2018	Brain	Bevacizumab and vorinostat	40	3	7.50	1	2
31	Ghiringhelli et al. [59]	2012	Colorectal	Bevacizumab and FOLFIRI-3 regimen (irinotecan, leucovorin and 5-fluorouracil)	49	1	2.04	0	1
32	Goss et al. [60]	2016	Lung	Osimertinib	210	1	0.48	0	1
33	Gronberg et al. [61]	2012	Lung	Enzastaurin	107	16	14.95	0	16
34	Guigay et al. [62]	2015	Head and Neck	Cetuximab, docetaxel and cisplatin	54	1	1.85	0	1
35	He et al. [63]	2020	Cervix	Cisplatin and paclitaxel chemotherapy with or without bevacizumab	264	24	9.09	0	24
36	Hirsch et al. [64]	2017	Lung	Onartuzumab, paclitaxel and carboplatin/cisplatin or placebo plus paclitaxel and carboplatin/cisplatin	109	3	2.75	0	3
37	Honecker et al. [65]	2013	Germ Cell	Cisplatin-based chemotherapy	193	4	2.07	0	4
38	Hu et al. [66]	2015	Lung	Nab-paclitaxel	56	12	21.43	4	8
39	Idelevich et al. [67]	2012	Oesophageal	Cisplatin, 5-FU, bevacizumab	28	3	10.71	0	3
40	Ikemura et al. [68]	2015	Lung	S-1 and irinotecan	31	1	3.23	0	1
41	Ishida et al. [69]	2015	Breast	Fulvestrant and trastuzumab (if HER2-positive disease)	117	1	0.85	0	1
42	Kakkos et al. [70]	2020	Mixed	Various	231	17	7.36	0	17
43	Karavasilis et al. [71]	2014	Lung	Erlotonib and docetaxel	50	1	2.00	0	1
44	Kim et al. [72]	2018	Lymph	Mogamulizumab or vorinostat	372	7	1.88	0	7
45	Kitayama et al. [73]	2017	Mixed	Various	97	29	29.90	0	29
46	Konecny et al. [74]	2015	Endometrial	Dovitinib	53	9	16.98	3	6
47	Kottschade et al. [75]	2013	Skin	Temozolomide and bevacizumab or nab-paclitaxel, carboplatin and bevacizumab	93	7	7.53	0	7
48	Lang et al. [76]	2012	Breast	Bevacizumab and capecitabine or paclitaxel	561	8	1.43	0	8
49	Lara et al. [77]	2016	Lung	Erlotinib or erlotinib plus carboplatin/paclitaxel	59	1	1.69	0	1
50	Larsen et al. [78]	2015	Gastric	Varied	129	21	16.28	0	21
51	Lee et al. [79]	2020	Ovary	Bevacizumab and sorafenib	54	5	9.26	0	5
52	Maio et al. [80]	2017	Mesothelium	Tremelimumab	571	17	2.98	0	17
53	Makielski et al. [81]	2015	Pancreas	Sorafenib and oxaliplatin and capecitabine	24	1	4.17	0	1
54	Matsumoto et al. [82]	2015	Ovary	Etoposide plus irinotecan	60	1	1.67	0	1
55	Michelsen & Sorensen [83]	2015	Lung	Platinum-vinorelbine plus bevacizumab with/without pemetrexed	42	10	23.81	0	10
56	Mountzios et al. [84]	2012	Lung	Bevacizumab and paclitaxel	30	1	3.33	0	1
57	Nagane et al. [85]	2012	Brain	Bevacizumab	31	1	3.23	0	1
58	Okines et al. [86]	2013	Gastric	Epirubicin, cisplatin and capecitabine plus bevacizumab	200	15	7.50	0	15
59	Ottosson et al. [87]	2020	Bladder	Varied	126	45	35.71	0	45
60	Peeters et al. [88]	2013	Colorectal	Trebananib and FOLFIRI	144	10	6.94	0	10
61	Pitz et al. [89]	2015	Brain	PX-866	33	2	6.06	0	2
62	Powell et al. [90]	2013	Lung	Topotecan	42	1	2.38	0	1
63	Ramos et al. [91]	2017	Bladder	Varied	1762	144	8.17	0	144
64	Reck et al. [92]	2014	Lung	Docetaxel plus nintedanib or docetaxel plus placebo	1314	3	0.23	0	3
65	Reyes-Botero et al. [93]	2018	Brain	Temozolomide plus bevacizumab	66	3	4.55	0	3
66	Rivera et al. [94]	2015	Gastric	Reduced dose docetaxel, oxaliplatin and capecitabine	43	4	9.30	0	4
67	Saad et al. [95]	2021	Prostate		982	15	1.53	0	15
68	Salinaro et al. [96]	2020	Ovary	Neoadjuvant	230	16	6.96	0	16
69	Salles et al. [97]	2020	Lymph	Tafasitamab and lenalidomide	156	7	4.49	2	5
70	Seidel et al. [98]	2012	Brain	Bevacizumab	2855	143	5.01	0	143
71	Slagter et al. [99,100]	2020	Gastric	Epirubicin, cisplatin, oxaliplatin and capecitabine	781	78	9.99	0	1
72	Sonpavde et al. [100]	2012	Prostate	Docetaxel plus prednisone with placebo or AT-101	220	9	4.09	0	9
73	Stevenson et al. [101]	2012	Lung	Bevacizumab plus pemetrexed and carboplatin followed by maintenance BVZ	40	3	7.50	0	3
74	Tahover et al. [102]	2015	Colorectal	Bevacizumab with other chemotherapies	308	20	6.49	0	20
75	Tan et al. [103]	2021	Breast	Pertuzumab and trastuzumab	500	4	0.80	0	4
76	Tryfonidis et al. [104]	2013	Breast	Docetaxel, epirubicin and bevacizumab	83	1	1.20	0	1
77	Tunio et al. [105]	2012	Renal	Thalidomide	80	7	8.75	0	7
78	Uetake et al. [106]	2015	Colorectal	mFOLFOX6 + bevacizumab	45	1	2.22	0	1
79	Usmani et al. [107]	2019	Blood	Pembrolizumab plus lenalidomide and dexamethasone	301	2	0.66	0	2
80	Vaishampayan et al. [108]	2014	Prostate	Bevacizumab and satraplatin in docetaxel-pretreated	31	2	6.45	0	2
81	Valle et al. [109]	2021	Liver	All patients received intravenous cisplatin 25 mg/m^2^ and gemcitabine 1000 mg/m^2^ (on days 1 and 8 in 21-day cycles), for a maximum of eight cycles + additional treatment	309	17	5.50	0	17
82	Wolff et al. [110]	2012	Colorectal	Enzastaurin with 5-FU/leucovorin plus bevacizumab	117	10	8.55	0	10
83	Yamazaki et al. [111]	2016	Colorectal	Bevacizumab + FOLFIRI or Bevacizumab + mFOLFOX6	395	26	6.58	0	26
84	Yardley et al. [112]	2012	Breast	Sunitinib	83	2	2.41	0	2
85	Zalcman et al. [113]	2016	Mesothelium	Bevacizumab, pemetrexed and cisplatin	448	15	3.35	0	15
86	Baggstrom et al. [114]	2017	Lung	Sunitinib after platinum-based chemotherapy	210	1	0.48	0	0
87	Chavan et al. [115]	2017	Ovary	Various	144	20	13.89	0	20
88	Duivenvoorden et al. [116]	2016	Bladder	Various	761	106	13.93	0	106
89	Gay et al. [117]	2012	Blood	Thalidomide or lenalidomide, and dexamethasone	411	49	11.92	0	49
90	Hong et al. [118]	2012	Colorectal	Bevacizumab plus doublet combination chemotherapy	76	1	1.32	0	1
91	Kang et al. [118]	2012	Gastric	Various	3095	103	3.33	0	103
92	Li et al. [119]	2017	Gastric	Modified FOLFOX6	39	3	7.69	0	3
93	Martella et al. [85]	2022	Blood	Various	222	50	22.52	0	50
94	Matikas et al. [120]	2016	Lung	Bevacizumab-containing chemotherapy treatments, in conjunction with paclitaxel/docetaxel/cisplatin/carboplatin	314	9	2.87	0	9
95	Monk et al. [121]	2018	Ovary	Paclitaxel and elesclomol sodium	56	7	12.50	5	2
96	Slavicek et al. [122]	2014	Colorectal	Various	3187	105	3.29	0	105
97	Tachihara et al. [123]	2020	Lung	Cisplatin-based adjuvant chemotherapy and pemetrexed	21	1	4.76	0	1
98	Tewari et al. [124]	2018	Cervix	Various regimens involving cisplatin/paclitaxel/topotecan/bevacizumab	452	22	4.87	0	22
99	Yildiz et al. [125]	2012	Colorectal	FOLFIRI and bevacizumab	332	4	1.20	0	4
100	Lee et al. [126]	2013	Colorectal	Bevacizumab and standard chemotherapy combinations	40	1	2.50	0	1
101	Reynes et al. [127]	2016	Brain	Temozolomide and irinotecan	27	1	3.70	0	1
102	Pinto et al. [128]	2021	Mesothelium	Gemcitabine with/without ramucirumab	165	20	12.12	15	5

Abbreviations: GI: gastrointestinal, VTE: venous thromboembolism, C: total number of patients, N: number of venous thromboembolism cases, P = crude prevalence of venous thromboembolism in individual studies.

**Table 3 diagnostics-12-02954-t003:** Study characteristics stratified by frequency, dosage, and duration of treatment regimen.

Study ID	Author	Year	Treatment Dose	Treatment Duration	Treatment Cycle Frequency
1	Affronti et al. [29]	2018	Bevacizumab (10 mg/kg IV) and Rilotumumab (20 mg/kg IV)	Bevacizumab (every 2 weeks for up to 12 cycles, with three infusions of Avastin every 2 weeks). Rilotumumab (every 2 weeks following the administration of Avastin for up to 12 cycles. Three infusions of Avastin at 10 mg/kg followed by rilotumumab at 20 mg/kg)	6 weeks
2	Alexander et al. [30]	2012	Thalidomide (200 mg daily drom Day 1 of radiation therapy, increasing by 100–200 to 1200 mg every 1–2 weeks until tumour progression or unacceptable toxicity)	N/R	N/R
3	Alvarez et al. [31]	2014	Bevacizumab (10 mg/kg IV every other week, e.g., day 1 and 15) plus temsirolimus (25 mg IV weekly, e.g., day 1, 8, 15 and 22) or a 4 week cycle	Until disease progression or adverse event prohibits further therapy	4 weeks
4	Assenat et al. [32]	2021a	Patients received AG [IV injection of nab-paclitaxel over 30 min followed by gemcitabine] at day 1, 8 and 15, while FFX was delivered at day 29 and 43 (IV injection of oxaliplatin for 2 h, irinotecan for 90 min and leucovorin for 2 h after 1 h rest, followed by fluorouracil bolus injection and then continuous 46 h infusion).	Median of 4 (1–9) cycles in 8.5 months (0.5–19.8 months)	N/R
5	Assenat et al. [33]	2021b	Patients received 1000 mg/m^2^ IV gemcitabine, 30 minutes infusion, on days 1, 8, 15, 22, 29, 36 and 43, during the first 8 weeks of treatment, then on days 1, 8 and 15, 3 weeks out of a 4-week cycle. They also received weekly IV trastuzumab, 4 mg/kg 90 min infusion on Day 1, 2 mg/kg on Days 8 and 15, 30 min infusion, and 100 mg/day erlotinib per os.	Median duration of 16.1 weeks	N/R
6	Bai et al. [34]	2015	mFOLFOX-6 (oxaliplatin 85 mg/m^2^ dL 5-FU bolus 400 mg/m^2^ d1, 5-FU 2400 mg/m^2^ continuous infusion for 46 h, every 2 weeks), XELOX (oxaliplatin 130 mg/m^2^ d1, capecitabine 2000 mg/m^2^ d1–14 every 3 weeks), or modified FOLFIRI (irinotecan 180 mg/m^2^ d1, 5-FU bolus 400 mg/m^2^ d1, 5FU 2400 mg/m^2^ continuous infusion for 46 h every 2 weeks), in combination with bevacizumab 5 mg/kg every 2 weeks (5-FU-based regimens) or 7.5 mg/kg every 3 weeks (capecitabine-based regimens).	N/R	N/R
7	Balar et al. [35]	2013	Patients initially received bevacizumab 10 mg/kg intravenously (IV) followed 2 weeks later with combination therapy. Gemcitabine 1000 mg/m^2^ on days 1 and 8 and carboplatin IV at area under the [concentration-time] curve (AUC) 5.0 on day 1 were administered every 21 days. Bevacizumab 15 mg/kg IV was administered on day 1 of each 21-day cycle	Median of 6 cycles administered	3 weeks
8	Basso et al. [36]	2013	PLD was administered at 20 mg/mq every two weeks for a maximum of 12 cycles.	Mean of 7.8 cycles	2 weeks
9	Bear et al. [37]	2015	Patients received one of three docetaxel-based neoadjuvant regimens for four cycles: docetaxel alone (100 mg/m^2^) with addition of capecitabine (825 mg/m^2^) oral twice daily days 1–14, 75 mg/m^2^) docetaxel) or with addition of gemcitabine (1000 mg/m^2^) days 1 and 8 intravenously, 75 mg/m^2^ docetaxel), all followed by neoadjuvant doxorubicin and cyclophosphamide (60 mg/m^2^) and 600 mg/m^2^) intravenously) every 3 weeks for four cycles. Those randomly assigned to bevacizumab groups were to receive bevacizumab (15 mg/kg, every 3 weeks for six cycles) with neoadjuvant chemotherapy and postoperatively for ten doses.	Various	Various
10	Buxo et al. [38]	2018	Carboplatin IV at an area under the curve of 5 mg/mL/min on day 1; cetuximab at an initial dose of 400 mg/m^2^ IV as a 2 h intravenous infusion, followed by 250 mg/m^2^ IV weekly as a 1 h infusion; and oral tegafur 500 mg/m^2^ every 12 h for 21 consecutive days	Median of 4.5 cycles, for 13.5 weeks	N/R
11	Campbell et al. [39]	2012	Administered orally once daily on days 1 through 28 of a 28-day cycle. Cediranib was initially dosed at 45 mg daily, but due to substantial rates of toxicity the protocol was amended in June 2007 to decrease the starting dose to 30 mg daily.	Median of 2 cycles, range 1–14	N/R
12	Chekerov et al. [40]	2018	Topotecan (1.25 mg/m^2^ on days 1–5) plus either oral sorafenib 400 mg or placebo twice daily on days 6–15	6 cycles	3 weeks
13	Chen et al. [41]	2015	One cycle of therapy consisted of oral vorinostat 200 mg twice daily for 14 days followed by a 7-day break, and intravenous rituximab 375 mg/m^2^ on day 1 of a 21-day cycle.	Median of 11.5 cycles, range 1–69, median duration is 17.8 months	N/R
14	Chibaudel et al. [42]	2019	Modified FOLFOX7 (5-FU/folinic acid and oxaliplatin) with aflibercept at 4 mg/kg every 2 weeks followed by maintenance therapy with fluoropyrimidine with aflibercept until disease progression or limiting toxicity.	6 cycles	2 weeks
15	Ciombor et al. [43]	2014	Bortezomib was administered at a dose of 1.3 mg/m^2^ IV push over 3–5 s on days 1, 4, 8, 11 of a 21-day cycle. Doxorubicin was administered at a dose of 15 mg/m^2^ IV over 5–15 min on days 1, 8 of each 21-day cycle. The first dose of doxorubicin was administered on day 8 of cycle 1 after the first two doses of bortezomib (cycle 1, day 8). On days when both bortezomib and doxorubicin were administered (days 1 and 8), doxorubicin was administered before bortezomib. Patients continued to receive chemotherapy until progression. Doxorubicin discontinued after receiving 12 cycles, regardless of response.	12 cycles maximum, median 3.8	3 weeks
16	Cremolini et al. [44]	2020	In the control group, patients received first-line mFOLFOX6 (85 mg/m^2^ of intravenous oxaliplatin concurrently with 200 mg/m^2^ of leucovorin over 120 min; 400 mg/m^2^ intravenous bolus of fluorouracil; 2400 mg/m^2^ continuous infusion of fluorouracil for 48 h) plus bevacizumab (5 mg/kg intravenously over 30 min) followed by FOLFIRI (180 mg/m^2^ of intravenous irinotecan over 120 min concurrently with 200 mg/m^2^ of leucovorin; 400 mg/m^2^ intravenous bolus of fluorouracil; 2400 mg/m^2^ continuous infusion of fluorouracil for 48 h) plus bevacizumab after disease progression. In the experimental group, patients received FOLFOXIRI (165 mg/m^2^ of intravenous irinotecan over 60 min; 85 mg/m^2^ intravenous oxaliplatin concurrently with 200 mg/m^2^ of leucovorin over 120 min; 3200 mg/m^2^ continuous infusion of fluorouracil for 48 h) plus bevacizumab followed by the reintroduction of the same regimen after disease progression.	Maximum 8 cycles	2 weeks
17	de Vos et al. [45]	2014	For Cycle 1, all patients were treated using an intra-patient dose-escalation schedule. 1 mg/kg Day 1, 2 mg/kg Day 4, 4 mg/kg Day 8, 8 thereafter. Subsequent cycles consisted of 4 doses of 8 mg/kg on Days 1, 8, 15, and 22. Patients were treated with 2 cycles after a complete remission (CR) or until disease progression for a maximum of 12 cycles.	up to 12 cycles	6 weeks
18	DeCensi et al. [46]	2019	5 mg/daily	3 years	Daily
19	Deschenes-Simard et al. [47]	2021	Various	N/R	N/R
20	Donskov et al. [48]	2018	IFN 3 MIU subcutaneously (SC) daily and IL2 2.4 MIU/m^2^ sc twice daily, 5 days per week for two consecutive weeks every 28-day-cycle, for 9 months; or supplemented with BEV 10 mg/kg, every 2 weeks intravenously (IV) until progression, unacceptable toxicity, or 1 year following no evidence of disease (NED)	9 months	4 weeks
21	Dowell et al. [49]	2012	Previously untreated MM patients with advanced, unresectable disease received cisplatin (75 mg/m^2^), pemetrexed (500 mg/m^2^), and bevacizumab (15 mg/kg) intravenously every 21 days for a maximum of 6 cycles. Patients with responsive or stable disease received bevacizumab (15 mg/kg) intravenously every 21 days until progression or intolerance.	Median of 6 cycles of chemotherapy	3 weeks
22	Dummer et al. [50]	2012	PLD 20 mg/m^2^ on days 1 and 15	Maximum 6 cycles	4 weeks
23	Duvic et al. [51]	2015	Brentuximab vedotin was administered intravenously at 1.8 mg/kg every 21 days for a maximum of eight doses. Patients with partial or stable response were eligible to receive up to eight additional doses. Patients with complete response could receive two additional doses. Patients with breakthrough lesions could receive 1.2 mg/kg every 2 weeks at the discretion of the principal investigator.	Maximum 8 cycles	3 weeks
24	Fehr et al. [52]	2020	Docetaxel 75 mg/m^2^ and cisplatin 75 mg/m^2^ (duration of cycle 3 weeks)	2 cycles	N/R
25	Feliu et al. [53]	2014	Intravenous bevacizumab 7.5 mg kg^−1^ and oxaliplatin 130 mg m^−2^ on day 1 of each cycle, plus oral capecitabine 1000 mg m^−2^ twice daily (BID) on days 1–14 of each cycle (patients with a baseline creatinine clearance of 30–50 mL min^−1^ had a 25% reduction in their initial capecitabine dose to 750 mg/m^2^ BID). Treatment was repeated every 3 weeks for 6 cycles. After 6 cycles, oxaliplatin was discontinued and patients continued to receive bevacizumab and capecitabine following the same regimen until progression or study discontinuation	Median of 6.8 months, range 0.2–25.2	3 weeks
26	Fleming et al. [54]	2014	Temsirolimus 25 mg IV weekly plus megestrol acetate 80 mg orally twice a day for 3 weeks alternating with tamoxifen 20 mg orally twice a day for 3 weeks	3 weeks	Twice a day
27	Folprecht et al. [55]	2016	MFOLFOX6 (5 mg/m^2^ oxaliplatin [2 (h) IV)] together with 350 mg/m^2^ leucovorin (2 h IV) followed by 5-FU (400 mg/m^2^ as bolus and 2400 mg/m^2^ IV over 46 h). Patients in the experimental arm received 4 mg/kg aflibercept (1 h IV) before chemotherapy.	The median number of aflibercept cycles was 7.0 (range 1–43). In the mFOLFOX6 and the aflibercept/mFOLFOX6 arms, the median number of oxaliplatin cycles was 10.0 (1–31) and 9.0 (1–40), and the median number of 5-FU cycles was 11.0 (1–43) and 10.0 (1–44), respectively. The median duration of exposure to aflibercept was 17.1 weeks (range 2–94). In the mFOLFOX6 and the aflibercept/mFOLFOX6 arms, the median duration of exposure to oxaliplatin was 23.2 (2–77) and 22.0 (2–84), and to 5-FU was 25.9 (2–95) and 24.1 (2–106) weeks, respectively.	N/R
28	Frizziero et al. [56]	2019	CarboEtop-1; etoposide 50 mg twice daily orally from day 1 to day 7 (inclusive) followed by carboplatin area under the curve (AUC) 5, intravenously on day 8, every 28 days; CarboEtop-2; etoposide 120 mg/m^2^ intravenously on days 1, 2, and 3, and carboplatin AUC 5 or 6 intravenously on day 1, every 21 days; CarboEtop-3; etoposide 100 mg/m^2^ intravenously on days 1, 2, and 3, and carboplatin AUC 4 or 5 intravenously on day 1, every 21 days; A higher proportion of patients received intravenous etoposide compared to oral etoposide, both in first-line (54.7% versus 45.3%) and second/third-line (58.8% versus 41.2%).	Median of 3.6 months, range 0.4–9.9	Various
29	Fuchs et al. [57]	2019	Temsirolimus 25 mg IV weekly plus megestrol acetate 80 mg orally twice a day for 3 weeks alternating with tamoxifen 20 mg orally twice a day for 3 weeks	3 weeks	Twice daily
30	Ghiaseddin et al. [58]	2018	Bevacizumab, 10 mg/kg IV every 2 weeks combined with VOR 400 mg PO daily for 7 days, then 7 days off in a 28-day cycle, vorinostat VOR 400 mg PO daily for 7 days, then 7 days off, in a 28-day cycle	N/R	4 weeks
31	Ghiringhelli et al. [59]	2012	Bevacizumab given at a dose of 5 mg/kg on day 1 every 2 weeks. FOLFIRI-3 regimen was given every 14 days as follows: on day 1, irinotecan 100 mg/m^2^ as a 1 h infusion, running concurrently with leucovorin 200 mg/m^2^ as a 2 h infusion via a Y-connector, followed by 5-FU 2000 mg/m^2^ as a 46 h infusion using an electric pump. On day 3, irinotecan 100 mg/m^2^ as a 1 h infusion was repeated, at the end of 5-FU infusion.	N/R	2 weeks
32	Goss et al. [60]	2016	Osimertinib 80 mg orally once daily	N/R	Daily
33	Gronberg et al. [61]	2012	Oral maintenance enzastaurin (1125 mg on Day 1 followed by 500 mg daily) or placebo	N/R	Daily
34	Guigay et al. [62]	2015	docetaxel 75 mg/m^2^ IV day 1, cisplatin 75 mg/m^2^ IV day 1, and cetuximab on days 1, 8, and 15 (400 mg/m^2^ IV day 1 of cycle 1 and 250 mg/m^2^ IV weekly on subsequent administrations)		4 weeks
35	He et al. [63]	2020	Intravenous chemotherapy regimen consisted of cisplatin (at a dose of 50 mg per square metre of body surface area) plus paclitaxel (at a dose of 175 mg/m^2^ on day 1); the intravenous BEV regimen was a dose of 15 mg/kg on day 1	N/R	3 weeks
36	Hirsch et al. [64]	2017	N/R	N/R	N/R
37	Honecker et al. [65]	2013	Carboplatin was applied either as single-agent adjuvant treatment for pure seminoma or combined with etoposide as high-dose first-salvage treatment after cisplatin-based chemotherapy. Cisplatin-based combination chemotherapy consisted of 2 cycles with etoposide and bleomycin (PEB) as adjuvant therapy in nonseminoma or of 3–4 cycles combined with etoposide plus bleomycin (PEB), etoposide plus ifosfamide (VIP), or ifosfamide plus paclitaxel (TIP) for metastatic disease.	N/R	N/R
38	Hu et al. [66]	2015	Nab-paclitaxel 100 mg/m^2^ (IV) on days 1, 8 and 15 of a 28-day cycle	up to 6 cycles	4 weeks
39	Idelevich et al. [67]	2012	Bevacizumab 7.5 mg/kg followed by cisplatin 80 mg/m^2^ infusion on day 1 followed by 5-FU 1000 mg/m^2^ as a 96 h continuous infusion on days 1–4, separated by a 3-week interval.	4 days per cycle	3 weeks
40	Ikemura et al. [68]	2015	Irinotecan was administered at 60 mg/m^2^ on Days 1 and 8. Oral S-1 was administered on Days 1–14 every 3 weeks at 80 mg/day for patients with a body surface area of <1.25 m^2^, 100 mg/day for patients with a body surface area of 1.25–1.5 m^2^ and 120 mg/day for patients with a body surface area of >1.5 m^2^	N/R	3 weeks
41	Ishida et al. [69]	2015	fulvestrant 500 mg as two 5-mL intramuscular injections, one in each buttock, on days 0, 14, and 28 and every 28 days thereafter.		4 weeks
42	Kakkos et al. [70]	2020	Various	N/R	N/R
43	Karavasilis et al. [71]	2014	Erlotinib for 12 consecutive days prior to docetaxel (Arm A) or after docetaxel (Arm B). Erlotinib was taken orally at a dose of 150 mg every day for 12 consecutive days and docetaxel was administered intravenously at a dose of 75 mg/m^2^.	N/R	3 weeks
44	Kim et al. [72]	2018	Mogamulizumab (1 mg/kg intravenously on a weekly basis for the first 28-day cycle, then on days 1 and 15 of subsequent cycles) or vorinostat (400 mg daily)	N/R	4 weeks
45	Kitayama et al. [73]	2017	Various	N/R	N/R
46	Konecny et al. [74]	2015	Dovitinib (500 mg per day, orally, on a 5 days-on and 2 days-off schedule	N/R	N/R
47	Kottschade et al. [75]	2013	Temozolomide (200 mg/m^2^ on d. 1–5) and bevacizumab (10 mg/kg IV d. 1 and 15) every 28 days (Regimen temozolomide/bevacizumab [TB]) or nab-paclitaxel (100 mg/m^2^ [80 mg/m^2^ post addendum 5-secondary to toxicity] days 1, 8 and 15), bevacizumab (10 mg/kg on days 1 and 15), and carboplatin (AUC 6 day 1 [AUC 5 post addendum 5]) every 28 days (Regimen ABC)	N/R	4 weeks
48	Lang et al. [76]	2012	Arm A: bevacizumab 10 mg/kg days 1 and 15; paclitaxel 90 mg/m^2^ days 1, 8, and 15, every 4 weeks; or Arm B: bevacizumab 15 mg/kg day 1; capecitabine 1000 mg/m^2^ BID, days 1–14, every 3 weeks, until disease progression, unacceptable toxicity or consent withdrawal.	Various	Various
49	Lara et al. [77]	2016	Erlotinib 150 mg orally daily (Arm 1) or erlotinib 150 mg orally daily on days 2–16 plus 4 cycles of carboplatin (AUC 5 day 1) and paclitaxel (200 mg/m^2^ IV day 1), followed by erlotinib 150 mg orally (Arm 2)	N/R	N/R
50	Larsen et al. [78]	2015	Various	N/R	N/R
51	Lee et al. [79]	2020	Bevacizumab (5 mg/kg IV every 2 weeks) was given with sorafenib 200 mg bid 5 days-on/2 days-off.	N/R	N/R
52	Maio et al. [80]	2017	Intravenous tremelimumab (10 mg/kg) or placebo every 4 weeks for 7 doses and every 12 weeks thereafter until a treatment discontinuation criterion was met.	N/R	Various
53	Makielski et al. [81]	2015	Sorafenib 200 mg orally twice daily along with oxaliplatin 85 mg/m^2^ IV on days 1 and 15, followed by capecitabine 2250 mg/m^2^ orally every 8 h for six doses starting on days 1 and 15 of a 28-day cycle	N/R	4 weeks
54	Matsumoto et al. [82]	2015	Oral etoposide (50 mg/m^2^ once a day) from day 1 to day 21 and IV irinotecan (70 mg/m^2^) on days 1 and 15	up to 6 cycles	4 weeks
55	Michelsen & Sorensen [83]	2015	Various	N/R	N/R
56	Mountzios et al. [84]	2012	Aclitaxel (90 mg/m^2^, days 1, 8 and 15) along with bevacizumab (10 mg per kg of body weight, days 1 and 15) in cycles of 28 days.	N/R	4 weeks
57	Nagane et al. [85]	2012	10 mg/kg bevacizumab as an intravenous infusion administered over 90 (±15) min on Day 1 of each cycle, which could be reduced to 30 min by Cycle 3 if no infusion reactions occurred.	N/R	N/R
58	Okines et al. [86]	2013	ECX comprises 3-weekly epirubicin 50 mg/m^2^ and cisplatin 60 mg/m^2^ IV (day 1), with capecitabine 1250 mg/m^2^/day (divided doses days 1–21), plus bevacizumab 7.5 mg/kg IV (day 1) added in the ECX-B arm. Surgery was scheduled 5 to 6 weeks after the last capecitabine dose of the third cycle and postoperative chemotherapy (three cycles) restarted 6–10 weeks after surgery. ECX-B patients then received six 3-weekly cycles of maintenance bevacizumab 7.5 mg/kg IV	N/R	N/R
59	Ottosson et al. [87]	2020	Various	N/R	N/R
60	Peeters et al. [88]	2013	Intravenous trebananib 10 mg kg^−1^ once weekly (QW) (Arm A) or placebo QW (Arm B)	N/R	Weekly
61	Pitz et al. [89]	2015	8 mg daily	N/R	8 weeks
62	Powell et al. [90]	2013	topotecan (4.0 mg/m^2^) on days 1, 8, and 15, and bevacizumab (10 mg/kg) on days 1 and 15 as intravenous infusions on a 28-day treatment cycle	N/R	4 weeks
63	Ramos et al. [91]	2017	Gemcitabine, cisplatin, nonplatinum regimens, etc.	N/R	N/R
64	Reck et al. [92]	2014	Nintedanib 200 mg orally twice daily or matching placebo on days 2–21	N/R	3 weeks
65	Reyes-Botero et al. [93]	2018	TMZ administered at 130–150 mg/m^2^ per day for 5 days every 4 weeks plus Bev administered at 10 mg/kg every 2 weeks	N/R	N/R
66	Rivera et al. [94]	2015	“miniDOX” regimen (D: 40 mg/m^2^ IV, day 1; O: 80 mg/m^2^ IV, day 1; C: 625 mg/m^2^ PO BID, day 1 to day 21, every 21 days; after six courses, only C was maintained)	N/R	3 weeks
67	Saad et al. [95]	2021	Oral apalutamide 240 mg once daily plus oral abiraterone acetate 1000 mg once daily and oral prednisone 5 mg twice daily (apalutamide plus abiraterone-prednisone group) or placebo plus abiraterone acetate and prednisone (abiraterone-prednisone group)	N/R	4 weeks
68	Salinaro et al. [96]	2020	Various	N/R	N/R
69	Salles et al. [97]	2020	Afasitamab was administered intravenously at a dose of 12 mg/kg, over approximately 2 h. For cycles 1–3, tafasitamab was administered weekly on days 1, 8, 15, and 22; an additional loading dose was administered on day 4 of cycle 1. From cycle 4, tafasitamab was administered every 14 days,17 on days 1 and 15 of each cycle. Patients self-administered lenalidomide capsules orally, starting with 25 mg daily on days 1–21 of each 28-day cycle. A stepwise dose reduction (decrease by 5 mg per day in each step, only once per cycle, without re-escalation) of lenalidomide was done in cases of protocol-defined toxicities.	N/R	4 weeks
70	Seidel et al. [98]	2012	Bevacizumab 5 mg/kg (n = 12) or 10 mg/kg (n = 34) every 2 weeks until disease progression or treatment-limiting toxicity	N/R	2 weeks
71	Slagter et al. [99,100]	2020	Epirubicin (50 mg/m^2^ on day 1), cisplatin (60 mg/m^2^ on day 1), or oxaliplatin (130 mg/m^2^ on day 1), and capecitabine (either 1000 mg/m^2^ twice daily on day 1–14 in combination with epirubicin and cisplatin or 625 mg/m^2^ twice daily on day 1–21 in combination with epirubicin and oxaliplatin) (ECC/EOC)	3 cycles	3 weeks
72	Sonpavde et al. [100]	2012	Docetaxel (75 mg/m^2^ day 1) and prednisone 5 mg orally twice daily every 21 days with either AT-101 (40 mg) or placebo twice daily orally on days 1–3	N/R	3 weeks
73	Stevenson et al. [101]	2012	Bevacizumab 15 mg/kg, pemetrexed 500 mg/m^2^ and carboplatin at an area under the concentration-time curve of 6 intravenously on day 1 every 21 days. Responding or stable patients who completed 6 cycles then received bevacizumab maintenance every 21 days until disease progression.	N/R	3 weeks
74	Tahover et al. [102]	2015	Bevacizumab was administered in combination with FOLFOX (modified FOLFOX6–oxaliplatin, leucovorin, 5-fluorouracil [5-FU]) in 40.3%, FOLFIRI (leucovorin, 5-FU, irinotecan) in 19.8%, FOLFOX-FOLFIRI/FOLFIRI-FOLFOX in sequence in 24.0%, CapeOx (oxaliplatin, capecitabine) in 6.5%, and 5-FU/LV or capecitabine monotherapy in 9.4%.	N/R	N/R
75	Tan et al. [103]	2021	Intravenous pertuzumab (840 mg loading dose, followed by 420 mg maintenance doses) plus intravenous trastuzumab (8 mg/kg loading dose, followed by 6 mg/kg maintenance doses) or the fixed-dose combination of pertuzumab and trastuzumab for subcutaneous injection (1200 mg pertuzumab plus 600 mg trastuzumab loading dose in 15 mL, followed by 600 mg pertuzumab plus 600 mg trastuzumab maintenance doses in 10 mL)	N/R	3 weeks
76	Tryfonidis et al. [104]	2013	received B (15 mg/kg), E (75 mg/m^2^) and D (75 mg/m^2^) with prophylactic G-CSF support every 3 weeks (q3w) for up to 9 cycles followed by B (15 mg/kg q3w) until disease progression	up to 9 cycles	3 weeks
77	Tunio et al. [105]	2012	Treatment consisted of gemcitabine in a 6 h infusion on days 1 and 8, and cisplatin at 75 mg/m on day 2 of a 3-week cycle. During phase I of the trial, the dose of gemcitabine was escalated from 130 to 170, 210 and 250 mg/m. In phase I of the trial, groups of six, seven, eight and eight patients were treated at the four dose levels of gemcitabine. In phase II, the remaining 32 patients all received gemcitabine at 250 mg/m.	N/R	Daily
78	Uetake et al. [106]	2015	On day 1, bevacizumab (5 mg/kg), levohorinate (200 mg/m^2^), 5-fluorouracil ([5-FU]; 400 mg/m^2^), and oxaliplatin (85 mg/m^2^) were rapidly injected intravenously, followed by a 46 h continuous intravenous infusion of 5-FU (2400 mg/m^2^). Each cycle of the treatment steps was repeated every 2 weeks.	N/R	2 weeks
79	Usmani et al. [107]	2019	Oral lenalidomide 25 mg on days 1–21 and oral dexamethasone 40 mg on days 1, 8, 15, and 22 every 4 weeks, with or without intravenous pembrolizumab 200 mg every 3 weeks	N/R	4 weeks
80	Vaishampayan et al. [108]	2014	Bevacizumab treatment was administered at 10 mg/kg intravenously on day 1, and 15 mg/kg on day 15, of each 35-day cycle. Premedications were allowed at the treating physician’s discretion. Satraplatin 80 mg/m^2^ was taken orally with fasting for 1 h prior, and 2 h after dosing. Prednisone 5 mg twice daily was taken with meal.	N/R	35 days
81	Valle et al. [109]	2021	Intravenous ramucirumab 8 mg/kg or placebo (on days 1 and 8 in 21-day cycles) or oral merestinib 80 mg or placebo (once daily) until disease progression, unacceptable toxicity, death, or patient or investigator request for discontinuation. All participants received intravenous cisplatin 25 mg/m^2^ and gemcitabine 1000 mg/m^2^ (on days 1 and 8 in 21-day cycles), for a maximum of eight cycles	Maximum 8 cycles	3 weeks
82	Wolff et al. [110]	2012	Both arms received LV5FU2 plus bevacizumab (Genentech/Roche, South San Francisco, CA, USA) on day 1 of each cycle (2 weeks): leucovorin 400 mg/m^2^ intravenously (IV), then 5-fluorouracil 400-mg/m^2^ bolus followed by 2400 mg/m^2^ IV over 46 h, and bevacizumab 5 mg/kg IV.	N/R	2 weeks
83	Yamazaki et al. [111]	2016	Bevacizumab (5 mg/kg) followed by FOLFIRI (irinotecan 150 mg/m^2^; l-leucovorin 200 mg/m^2^; intravenous bolus of fluorouracil 400 mg/m^2^, continuous infusion of fluorouracil 2400 mg/m^2^), or bevacizumab followed by mFOLFOX6 (oxaliplatin 85 mg/m^2^ instead of irinotecan)	N/R	2 weeks
84	Yardley et al. [112]	2012	Sunitinib monotherapy at a starting dose of 37.5 mg orally on a continuous daily dosing schedule; one treatment cycle was considered to be 4 weeks.	N/R	4 weeks
85	Zalcman et al. [113]	2016	Intravenously 500 mg/m^2^ pemetrexed plus 75 mg/m^2^ cisplatin with (PCB) or without (PC) 15 mg/kg bevacizumab	Maximum 6 cycles	3 weeks
86	Baggstrom et al. [114]	2017	Patients received maintenance sunitinib, 37.5 mg/d continuously, or placebo until disease progression or intolerable toxicity.	N/R	N/R
87	Chavan et al. [115]	2017	Various	Various	Various
88	Duivenvoorden et al. [116]	2016	Various	Various	Various
89	Gay et al. [117]	2012	Thalidomide was given at a dose ranging from 100 mg/day to 400 mg/day continuously; lenalidomide dose was 25 mg/day, days 1 to 21 on a 28-day cycle. All patients received dexamethasone, either at high dose (40 mg orally on days 1–4, 9–12, and 17–20) or at low dose (40 mg orally on days 1, 8, 15, and 22).	N/R	4 weeks
90	Hong et al. [118]	2012	FOLFOX or FOLFIRI consisted of leucovorin 200 mg/m^2^ on day 1, 5-FU 400 mg/m^2^ bolus infusion on day 1, and 5-FU 2400 mg/m^2^ continuous infusion for 46 h, either with oxaliplatin 85 mg/m^2^ or with irinotecan 150 or 180 mg/m^2^ on day 1, respectively, and repeated every 2 weeks. CapeOX consisted of capecitabine 1000 mg/m^2^ twice daily on days 1–14 and oxaliplatin 130 mg/m^2^ on day 1 and again every 3 weeks.	N/R	N/R
91	Kang et al. [118]	2012	Various	Various	Various
92	Li et al. [119]	2017	mFOLFOX6 (leucovorin 400 mg/m, fluorouracil 400 mg/m bolus and 2400 mg/m continuous infusion over 46 h, oxaliplatin 85 mg/m) and bevacizumab (10 mg/kg) every 2 weeks until disease progression or intolerance.	Median 12 cycles, range 4–86	2 weeks
93	Martella et al. [85]	2022	Various	Various	Various
94	Matikas et al. [120]	2016	Various	Various	Various
95	Monk et al. [121]	2018	Paclitaxel 80 mg/m^2^ and elesclomol sodium 200 mg/m^2^ (equivalent of free elesclomol) were administered as two separate 1 h IV infusions weekly × 3 with a one-week rest	Various	weeks
96	Slavicek et al. [122]	2014	Various	Various	Various
97	Tachihara et al. [123]	2020	Adjuvant chemotherapy with four cycles of cisplatin-based treatment (75 mg/m^2^) plus pemetrexed (500 mg/m^2^) with vitamin supplementation every three weeks.	N/R	N/R
98	Tewari et al. [124]	2018	Various	Various	Various
99	Yildiz et al. [125]	2012	Various	Various	Various
100	Lee et al. [126]	2013	Eligible patients received bevacizumab (Avastin, Roche Products Ltd.), plus standard 5-fluoropyrimidine (5-FU)-based chemotherapy per physician’s choice (single-agent 5-FU or 5-FU plus oxaliplatin or irinotecan) until disease progression, unacceptable toxicity or death. The bevacizumab dose was fixed at 5 mg/kg every 2 weeks.	Various	Various
101	Reynes et al. [127]	2016	All patients received oral TMZ at a fixed and continuous dose of 50 mg/m^2^ divided into three daily intakes, except for a single 100 mg/m^2^ dose, administered between 3 and 6 h before every irinotecan infusion. Irinotecan was given intravenously at the previously established dose of 100 mg/m^2^ on days 8 and 22 of 28-day cycles.	N/R	4 weeks
102	Pinto et al. [128]	2021	Patients received intravenous gemcitabine 1000 mg/m^2^ on days 1 and 8 every 3 weeks, combined with either intravenous ramucirumab 10 mg/kg or matching placebo on day 1 of a 3-week cycle, until progressive disease, unacceptable toxicity, or withdrawal of consent to treatment occurred.	N/R	3 weeks

Abbreviations: N/R: not reported; IV: intravenous; SC: subcutaneous; CR: complete remission; NED: no evidence of disease; 5-FU: 5-fluoropyrimidine; PO: medication taken orally; BID: medication taken twice a day; FOLFOX: 5-fluorouracil/leucovorin combined with oxaliplatin; FOLFIRI: 5-fluorouracil/leucovorin combined with irinotecan; TMZ: Temozolomide; CapeOx: oxaliplatin and capecitabine.

**Table 4 diagnostics-12-02954-t004:** Pooled and crude prevalence of venous thromboembolism across various cancer phenotypes.

Cancer Phenotype	Number of Studies (N)	Total Number of Patients (n)	Crude Prevalence Rate	Pooled Prevalence Rate (Derived from Meta-Analysis)	95% CI	z-Score	*p*-Value
Overall	102	30671	5.78%	6%	0.06–0.07	18.53	<0.001
Cancer Phenotype							
Bladder	4	2700	11.30%	18%	0.10–0.28	6.53	<0.001
Blood	3	934	10.81%	N/A	N/A	N/A	N/A
Brain	8	3177	5.19%	4%	0.04–0.05	17.72	<0.001
Breast	8	3082	1.88%	1%	0.00–0.03	4.17	<0.001
Cervical	2	716	6.42%	N/A	N/A	N/A	N/A
Colorectal	15	5891	4.69%	5%	0.03–0.07	8.16	<0.001
Endometrial	3	173	11.56%	N/A	N/A	N/A	N/A
Gastric	7	4932	6.55%	9%	0.05–0.15	5.89	<0.001
Germ Cell	1	193	2.07%	N/A	N/A	N/A	N/A
Head and Neck	2	158	1.27%	N/A	N/A	N/A	N/A
Liver	2	347	5.19%	N/A	N/A	N/A	N/A
Lung	16	3228	3.97%	5%	0.02–0.09	4.32	<0.001
Lymph	6	699	3.58%	4%	0.02–0.07	4.69	0.05
Mesothelial	5	1286	4.82%	6%	0.03–0.11	5.24	<0.001
Mixed	2	328	14%	N/A	N/A	N/A	N/A
Neuroendocrine	1	113	6.19%	N/A	N/A	N/A	N/A
Oesophageal	2	328	9.76%	N/A	N/A	N/A	N/A
Ovarian	6	718	8.22%	8%	0.05–0.12	7.47	0.02
Pancreatic	3	144	28.47%	N/A	N/A	N/A	N/A
Prostate	3	1233	2.11%	N/A	N/A	N/A	N/A
Renal	2	198	11.11%	N/A	N/A	N/A	N/A
Skin	1	93	7.53%	N/A	N/A	N/A	N/A

Abbreviations: CI: confidence interval, N: number of studies, n: number of patients. Note: N/A = could not be generated as meta-analysis could not be performed due to limited number of studies (minimum of four studies required).

## Data Availability

The original contributions presented in the study are included in the article, and further inquiries can be directed to the corresponding author.

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
