# Peer review of "Venous Thromboembolism in Cancer Patients Undergoing Chemotherapy: A Systematic Review and Meta-Analysis"

_diagnostics, 2022, doi:10.3390/diagnostics12122954_

Round 1

Reviewer 1 Report

General comments:

The authors of this manuscript have performed a meta-analysis of all cancer studies with patients undergoing chemotherapy and/or radiotherapy reporting VTE.  The objective is to estimate the overall incidence of VTE in these patients with a view to improving patient care?  Although this may seem admirable, the authors seem unaware of recent guidelines which recommend stratifying cancer patients according to their VTE risk prior to the start of  chemotherapy (DOI: 10.1182/bloodadvances.2020003442).   using the Khorana risk score, the major determinant of which is cancer type.  Hence the clinical relevance of this type of analysis is doubtful.  As a general comment, a manuscript of 138 pages is unlikely to be suitable for publication.  I would recommend that the authors take a more focussed approach both to their study and to the discussion of their results in future submissions.

I have the following specific comments

1.     The title of the article and the study indicates that the authors propose to study the role of chemo-radiation on VTE risk however only 17 of the 208 studies included were of patients undergoing chemoradiation.  In the introduction, the authors comment on previous studies looking at VTE and chemoradiation, the studies they refer to are all chemotherapy studies.

2.    The major issue with the study methodology is the heterogeneity of the studies, it seems that any study or randomised trial of chemotherapy which may have reported VTE as a side effect is included, and I miss many studies which were particularly focussed on determining VTE risk is selected groups (eg Khorana et al 2013;119:648-655. , Mulder et al, doi: 10.1002/bjs.11665 and many more including some which also focussed on mixed groups.  I am not sure therefore if the search strategy was effective in answering the author’s question.  It is also unclear whether all VTE was captured as the aim of the studies included was not to determine VTE incidence.

3.    The date range for inclusion is also an issue, studies between 1973 and June 2022 were included.  The massive changes in cancer treatment and also in the diagnosis treatment and prevention of cancer associated VTE make this an unworkable date range.  The maximum date range, I would recommend in this field for such a study is 10 years.

4.    There is a heterogeneity also in the data available, it appears that some studies to not have the complete data required, I would suggest that the authors focuss only on the studies with complete data.

5.    No information is presented with regard to prophylaxis for VTE or anticoagulant use for other indications such as Atrial Fibrillation. This may have modified VTE risk in certain groups and indeed the use of prophylaxis has changed dramatically within the time period of the study.

6.    The length of the manuscript is a major issue, the authors state their aim is to estimate the overall risk of VTE however in both the results and discussion section they delineate and comment on the risk of each specific cancer type which seems to contradict the overall aim of the study, it also results in an overlong manuscript.

7.    In the discussion, the authors also comment on possible mechanisms for VTE in each cancer type, I would suggest they limit their comments to an overall discussion of contributing factors.

8.    No mention is made of the two landmark studies which showed that VTE prophylaxis with DOACS following risk assessment lowered the incidence of VTE during chemotherapy- resulting in a change in the guidelines.  N Engl J Med 2019; 380: 711-719; N Engl J Med 2019; 380: 720-728.

Author Response

At the outset, we would like to thank the reviewer for their comments and review of our work. We have made best of our efforts to improve the manuscript. A point-by-point rebuttal is provided below.

General comments:

C#1: The authors of this manuscript have performed a meta-analysis of all cancer studies with patients undergoing chemotherapy and/or radiotherapy reporting VTE.  The objective is to estimate the overall incidence of VTE in these patients with a view to improving patient care?  Although this may seem admirable, the authors seem unaware of recent guidelines which recommend stratifying cancer patients according to their VTE risk prior to the start of  chemotherapy (DOI: 10.1182/bloodadvances.2020003442).   using the Khorana risk score, the major determinant of which is cancer type.  Hence the clinical relevance of this type of analysis is doubtful.  As a general comment, a manuscript of 138 pages is unlikely to be suitable for publication.  I would recommend that the authors take a more focussed approach both to their study and to the discussion of their results in future submissions.

Reply# We thank the reviewer for comprehensive review of our work and comments that have helped us to significantly improve the clarity of, and analyses, in this manuscript. Indeed, the objective is to estimate the overall incidence of VTE with a view to improving patient care, but also to provide quantum of pooled prevalence which may be useful to help build awareness and inform patients as well as clinicians to have a quantitative understanding of VTE prevalence rates.

We thank the reviewer for the comment/s. We have now added the discussion on American Society of Hematology Guidelines. We have added the following statements to qualify this. We have also taken a more focussed approach and a fresh search was performed with a revised criterion to estimate prevalence of VTE in cancer patients receiving chemotherapy only.

Page 3/Introduction

Recent guidelines from American Society of Haematology published in early 2021 recommend stratifying cancer patients according to their VTE risk prior to the start of chemotherapy, as well as patient-specific factors, using the Khorana risk score, the major determinant of which is cancer phenotype [1]. This comes in the background of two landmark randomized clinical trials (RCTs), resulting in the change of guidelines, demonstrating VTE prophylaxis with direct oral anticoagulants (DOACs) following risk assessment lowered the incidence of VTE during chemotherapy [2-4]. Several societies or health systems beyond United States are yet to adopt these recommendations; besides, unwarranted variations in clinical care as well as poor adherence to recommendations or guidance vis a vis VTE risk assessment and optimal administration of thromboprophylaxis pose an ongoing real-world or systems challenge [5,6]. Moreover, literature is sparse when comparing the relative risk and prevalence of VTE across multiple cancer phenotypes – with studies only revealing VTE prevalence specific to a cancer phenotype and risk in homogenous cancer populations, vis a vis their ethnicity and treatment received. Understanding of, and estimates of, the pooled prevalence may also be useful to increase awareness on VTE risks in cancer patients undergoing chemotherapy as well to inform clinicians and patients on the quantum of the VTE prevalence/risks in cancer or across various types of cancer. This meta-analysis sought to investigate the pooled prevalence of venous thromboembolism in cancer patients receiving chemotherapy. There is also a gap in clinician knowledge pertaining to the specific risk that cancer phenotypes and chemotherapy poses to cancer patients. We have sought to address two key underlying questions through this meta-analysis:

1) what is the prevalence of VTE in cancer patients receiving chemotherapy? 

2) what is the prevalence of VTE stratified by cancer phenotype in patients undergoing chemotherapy?

I have the following specific comments

C#2: 1.     The title of the article and the study indicates that the authors propose to study the role of chemo-radiation on VTE risk however only 17 of the 208 studies included were of patients undergoing chemoradiation.  In the introduction, the authors comment on previous studies looking at VTE and chemoradiation, the studies they refer to are all chemotherapy studies.

Reply# We thank the reviewer for the suggestion. We have now revised the study to only include studies on VTE in cancer patients undergoing chemotherapy.

C#3: 2.    The major issue with the study methodology is the heterogeneity of the studies, it seems that any study or randomised trial of chemotherapy which may have reported VTE as a side effect is included, and I miss many studies which were particularly focussed on determining VTE risk is selected groups (eg Khorana et al 2013;119:648-655. , Mulder et al, doi: 10.1002/bjs.11665 and many more including some which also focussed on mixed groups.  I am not sure therefore if the search strategy was effective in answering the author’s question.  It is also unclear whether all VTE was captured as the aim of the studies included was not to determine VTE incidence.

Reply# We have re-ran the search and analysis with a new strategy focussed on studies, on VTE in cancer patients undergoing chemotherapy, published between 2012 and October 2022. Besides, additional studies were also included through handsearching of references from included studies as well as from other sources such as Google Scholar and ResearchGate. We also set out the inclusion criterion so as to include patients who are not on prophylactic anticoagulation concomitant to chemotherapy. Mulder paper was not included it had a subgroup of patients on prophylactic anticoagulation concomitant to chemotherapy. Khorana study included patients on radiation and chemotherapy and separate data on chemotherapy vs radiation therapy were not available. This was done to understand VTE risks in hospital setting and exclude influence of concomitant anticoagulation on VTE risks. A total of 2, 643 and 85 studies were identified from PubMed and other sources, respectively. On screening of 2723 titles/abstracts, 2172 studies were excluded. Out of these, full texts of 551 studies were assessed for eligibility. Overall, 102 studies were included in the final synthesis.  The current meta-analysis included 102 studies, which reported on VTE prevalence within cancer patients undergoing chemotherapy, without concomitant prophylactic anticoagulation, with a cumulative cohort of 30, 671 patients (1,773 with VTE, 28, 898 without). This has also been included in the Results.

C#4: 3.    The date range for inclusion is also an issue, studies between 1973 and June 2022 were included.  The massive changes in cancer treatment and also in the diagnosis treatment and prevention of cancer associated VTE make this an unworkable date range.  The maximum date range, I would recommend in this field for such a study is 10 years.

Reply: We thank the reviewer for the valuable suggestion. As per the recommendation, as indicated above, only articles published between 2012 and October 2022 were included in the current meta-analysis.

C#5: 4.    There is a heterogeneity also in the data available, it appears that some studies to not have the complete data required, I would suggest that the authors focus only on the studies with complete data.

Reply: We acknowledge the observation by the reviewer. For matters of consistency, since our main goal was to estimate VTE prevalence in cancer and across cancer phenotypes, all studies reporting on VTE in cancer, without prophylactic anticoagulation concomitant to chemotherapy, were included.

C#6: 5.    No information is presented with regard to prophylaxis for VTE or anticoagulant use for other indications such as Atrial Fibrillation. This may have modified VTE risk in certain groups and indeed the use of prophylaxis has changed dramatically within the time period of the study.

Reply# We thank the reviewer for the comment. We only included studies where cancer patients were not on prophylactic anticoagulation concomitant to chemotherapy. Although, this indeed is an important clinical consideration, especially in certain groups such as atrial fibrillation, this was outside the scope of our current study. However, we have included the following statement in the limitations.

Page 14 (Limitations)

       Additionally, documentation on certain groups such as atrial fibrillation and VTE recurrence were not available across all studies.

C#7: 6.    The length of the manuscript is a major issue, the authors state their aim is to estimate the overall risk of VTE however in both the results and discussion section they delineate and comment on the risk of each specific cancer type which seems to contradict the overall aim of the study, it also results in an overlong manuscript.

Reply# We thank the reviewer for the comment. We have now removed the discussion on each specific cancer type. We have made best of efforts to reduce the overall length of the manuscript.

C#8: 7.    In the discussion, the authors also comment on possible mechanisms for VTE in each cancer type, I would suggest they limit their comments to an overall discussion of contributing factors.

Reply# We have removed the discussion on individual cancer type and possible mechanisms for VTE. Instead, the current work focussed on overall discussion of contributing factors.

C#9: 8.    No mention is made of the two landmark studies which showed that VTE prophylaxis with DOACS following risk assessment lowered the incidence of VTE during chemotherapy- resulting in a change in the guidelines.  N Engl J Med 2019; 380: 711-719; N Engl J Med 2019; 380: 720-728.

Reply# We have included both these studies in the Introduction, as below.

       This comes in the background of two landmark randomized clinical trials (RCTs), resulting in the change of guidelines, demonstrating VTE prophylaxis with direct oral anticoagulants (DOACs) following risk assessment lowered the incidence of VTE during chemotherapy [2-4].

Reviewer 2 Report

Dear Authors, the paper is well written. The authors try to define incidence of VTE in in Cancer Patients Undergoing Chemotherapy and Radiotherapy

Thank you very much to the authors for this systematic review and Meta-Analysis, I found it very interesting. I appreciated the idea of the paper, but there are some points that need to be clarified:

1.       The title should include:  a systematic review and Meta-Analysis, not only the latest one

2.       Your question is not built on the PICO criteria, please specified what is P, I, C, O in your question

3.       “This study was registered in Open Science, registration number is “bqrjk”” Please quote the Uniform Resource Locator (URL)of the site I didn’t find this site and the registration on internet. Moreover, why didn’t you submit your Meta-Analysis to PROSPERO that is the best known site for Meta-Analysis? https://www.crd.york.ac.uk/PROSPERO/

4.       I review the Search Strategy used. Your paper is on venous thrombosis, but you use in your strategy many arterial thrombosis, these are completely different. Please remove this key word and verify your prisma research and results or change the title (“ Arterial and Venous thrombosis in Cancer Patients Undergoing Chemotherapy and Radiotherapy: A Systematic review and Meta-Analysis”) and the rest of the paper (introduction, results and limitations)

5.       Table 1 organize study in order by Author name. It will be better to use order of year of publication.

6.       Table 2 and table 3 organize studies in order by Author name, but are focus on treatment regimen. Please use the treatment regimen to organize the table.

7.       Figure 2 and 4 are a joke? Nobody can read this figure. Split the images in more pages

8.       You report the result as percentage, but CI is absolute, please decide how report the result in the test, if percentage also CI should be percentage.

9.       Chemotherapy radically change during different years; your oldest quoted paper was published in 1994. Can you perform a sub-analysis basing on time of publication (at least every 10 years, better every 5 years)?

10.   “These include tissue factor, microparticles, plasminogen activator inhibitor-1, cancer procoagulant, mucin, tumour-derived platelet agonists and inflammatory cytokines such as IL-6, IL-8 and IL-10” Please quote this novel paper on microparticles: “Gidaro A, Manetti R, Delitala AP, Soloski MJ, Lambertenghi Deliliers G, Castro D, Soldini D, Castelli R. Incidence of Venous Thromboembolism in Multiple Myeloma Patients across Different Regimens: Role of Procoagulant Microparticles and Cytokine Release. J Clin Med. 2022 May 11;11(10):2720. doi: 10.3390/jcm11102720. PMID: 35628848; PMCID: PMC9143530.”

11.   Divide limits section from conclusion that need a paragraph alone.

Author Response

At the outset, we would like to thank the reviewer for their comments and review of our work. We have made the best of our efforts to improve the manuscript. A point-by-point rebuttal is provided below.

C# Dear Authors, the paper is well written. The authors try to define incidence of VTE in in Cancer Patients Undergoing Chemotherapy and Radiotherapy

Thank you very much to the authors for this systematic review and Meta-Analysis, I found it very interesting. I appreciated the idea of the paper, but there are some points that need to be clarified:

Reply# We thank the reviewer for favorable review of our work. We have made a significant revision as per the comments/recommendations. 

  1. The title should include: a systematic review and Meta-Analysis, not only the latest one

Reply# We have included this in the title as per the suggestion.

  1. Your question is not built on the PICO criteria, please specified what is P, I, C, O in your question

Reply# Given that this is a meta-analysis on pooled prevalence, we didn't have a comparator arm in all studies. Our aim was to estimate the pooled prevalence of VTE in cancer patients undergoing chemotherapy.

  1. “This study was registered in Open Science, registration number is “bqrjk”” Please quote the Uniform Resource Locator (URL)of the site I didn’t find this site and the registration on internet. Moreover, why didn’t you submit your Meta-Analysis to PROSPERO that is the best known site for Meta-Analysis? https://www.crd.york.ac.uk/PROSPERO/

Reply# The URL of the revised protocol is https://osf.io/yn5br/. The PROSPERO registration has been riddled with inadvertent delays in publishing review registration (see Puljak et al BMJ Evid Based Med. 2021 https://pubmed.ncbi.nlm.nih.gov/32665222/). That's Open science was our platform for registration. The following statement has been updated.

This study was registered in Open Science, registration number is “yn5br” (https://osf.io/yn5br/).

  1. I review the Search Strategy used. Your paper is on venous thrombosis, but you use in your strategy many arterial thrombosis, these are completely different. Please remove this key word and verify your prisma research and results or change the title (“ Arterial and Venous thrombosis in Cancer Patients Undergoing Chemotherapy and Radiotherapy: A Systematic review and Meta-Analysis”) and the rest of the paper (introduction, results and limitations)

Reply# We have revised the search strategy and also the main document to focus on venous thromboembolism. The title also have been accordingly changed.

  1. Table 1 organize study in order by Author name. It will be better to use order of year of publication. The title also have been accordingly changed.

Table 2 and table 3 organize studies in order by Author name, but are focus on treatment regimen. Please use the treatment regimen to organize the table.

Reply# We have organized all three tables 1, 2, and 3 according to Study ID. Given a relatively large number of studies, reorganizing it according to year or treatment regimen will be time intensive. We thank the reviewer for the suggestion.

  1. Figure 2 and 4 are a joke? Nobody can read this figure. Split the images in more pages

Reply# The figures have now been split. See Figures 2 and 3.

  1. You report the result as percentage, but CI is absolute, please decide how report the result in the test, if percentage also CI should be percentage.

Reply# The results are now presented in percentages.

  1. Chemotherapy radically change during different years; your oldest quoted paper was published in 1994. Can you perform a sub-analysis basing on time of publication (at least every 10 years, better every 5 years)?

Reply# We thank the reviewer for the suggestion. We have revised the time period and ran the search strategy for the period 2012 to October 2022. A fresh analysis was performed.

  1. “These include tissue factor, microparticles, plasminogen activator inhibitor-1, cancer procoagulant, mucin, tumor-derived platelet agonists, and inflammatory cytokines such as IL-6, IL-8 and IL-10” Please quote this novel paper on microparticles: “Gidaro A, Manetti R, Delitala AP, Soloski MJ, Lambertenghi Deliliers G, Castro D, Soldini D, Castelli R. Incidence of Venous Thromboembolism in Multiple Myeloma Patients across Different Regimens: Role of Procoagulant Microparticles and Cytokine Release. J Clin Med. 2022 May 11;11(10):2720. doi: 10.3390/jcm11102720. PMID: 35628848; PMCID: PMC9143530.”

Reply# We have added this reference as suggested.

  1. Divide limits section from conclusion that need a paragraph alone.

Reply# The conclusions has now been separated from the Limitations section. We have also revised the Conclusion as below.

Page 14-15

In conclusion, this meta-analysis demonstrated a pooled prevalence estimate of 6%, with a range of 5 to 7%, of VTEs amongst cancer patients undergoing chemotherapy. Our study indicates there is substantial risk of developing VTE as a cancer patient on chemotherapy showing a compelling need for robust screening and subsequent prophylactic management to prevent future VTE. More efforts should be undertaken to implement adherence of American Society of Haematology guidelines on VTE risks and management in cancer patients undergoing chemotherapy [19].

Round 2

Reviewer 1 Report

The authors have addressed all my comments, I have no further comments to make on the revised manuscript.

Reviewer 2 Report

Dear authors

After this improvement of the manuscript, the article is almost fit for publication.

I suggest only a minor improvement:

Figure 2 and figure 3 are not readable (especially figure 3). divide them over several pages